# Supported storytelling through the 'Life Threads' approach for family members after traumatic brain injury: *"We've been through all of this trauma, and you're giving me some string?"*

Charlotte Jane Whiffin[1,2]*, Caroline Ellis-Hill[3], Alyson Norman[4], Morag Lee[5], Parmjeet Kaur Singh[5], Mark Holloway[6], Jo Clark-Wilson[6], Natasha Yasmin[1], Sara Rose[1], David Sheffield[1], Fergus Gracey[7]

**1** College of Health, and Humanities, University of Derby, Derby, United Kingdom, **2** Department of Clinical Neurosciences, University of Cambridge, Cambridge, United Kingdom, **3** Faculty of Health and Social Sciences, Bournemouth University, Bournemouth, United Kingdom, **4** School of Psychology, Faculty of Health and Human Sciences, University of Plymouth, Plymouth, United Kingdom, **5** Patient and Public Involvement Co-applicant, Derby, United Kingdom, **6** Head First (Assessment, Rehabilitation & Case Management) Ltd, Cranbrook, United Kingdom, **7** Faculty of Medicine and Health Sciences, University of East Anglia, Norwich, United Kingdom

* c.whiffin@derby.ac.uk

## Abstract

Traumatic brain injury (TBI) brings inevitable and significant changes for family members, yet there is little to relieve their trauma, resolve their grief, or prevent ongoing suffering. The aim of this qualitative pre-feasibility study was to understand the clinical potential of storytelling for families after TBI using the 'Life Threads' approach (LTA). An in-depth inductive qualitative design was adopted within an interpretivist paradigm. Following informed consent, participants took part in an online focus group, then engaged with the LTA over four weeks before completing an unstructured, in-person interview. A final focus group explored participants' reflections on the LTA. A purposive sample of 20 family members began the study, three withdrew after the first focus group leaving a final sample size of 17. The analytical approach used was Braun and Clarke's reflexive thematic analysis. Eleven participants reported tangible benefits from engaging with the LTA, and thirteen described being able to tell their story in a way not possible through traditional methods. Four main themes were identified: 'Scaling Cliffs with Broken Wings' and 'An Entanglement of Wellbeing' illustrated the evolving and contextual needs of families post-TBI. 'Hear Me, See Me: The Power of Story' highlighted how the LTA facilitated voice, agency, and sense-making, while 'Creating the Conditions for Stories to Be Told' identified the enabling environment required for such benefits to emerge. The findings suggest that the LTA offers a meaningful way for family members to explore, express, and make sense of their experiences following TBI. It supports narrative reconstruction, fosters connection and agency, and provides a rare opportunity for self-reflection. However,

**Data availability statement:** The data underlying this study cannot be made publicly available due to ethical restrictions. The dataset includes qualitative interview transcripts containing potentially identifiable and sensitive personal information. Although participants consented to the use of anonymised verbatim excerpts in publications, they did not consent to full transcript data being shared in the public domain. Furthermore, due to the nature of the data, complete anonymisation cannot be guaranteed, as participants may be identifiable through unique experiences described. These restrictions are in accordance with the study's approved ethical protocol. Data may be made available to future researchers who meet the criteria for access to confidential data, subject to review and approval by the study sponsor. Requests for data access should be directed to the sponsor at: nuhnt.researchsponsor@nhs.net.

**Funding:** This project is funded by the National Institute for Health and Care Research (NIHR) under its Research for Patient Benefit (RfPB) Programme (Grant Reference Number NIHR204092). The views expressed are those of the author(s) and not necessarily those of the NIHR or the Department of Health and Social Care.

**Competing interests:** I have read the journal's policy and the authors of this manuscript have the following competing interests: CJW and AN are co-chairs of Anchor Point, a national organisation driving change for families after Acquired Brain Injury (anchorpointabi.org); JCW, NY, MH, SR and FG are also working group members.

implementation must be trauma-informed, paced, and supported by skilled facilitators capable of holding space for complex emotional responses.

## Introduction

Traumatic brain injury (TBI) is a major cause of disability in the UK [1–3] and a global public health concern [4]. While survivors face a range of physical, cognitive, and emotional challenges, families also experience significant psychological impacts, including depression, anxiety, and strained relationships [5–7]. Family functioning can become disrupted, leading to further emotional distress for family members and poorer outcomes for the injured person [8–11]. Although caregiving can be physically demanding, changes in personality and cognition often present the greatest challenge to families post-TBI [12,13]. These effects may be long-lasting and the impact of TBI on families enduring. Family members themselves change in response to a major life event, affecting their mental health and quality of life [14].

Support for families has been recognised as inconsistent, inefficient and often inadequate, with families struggling to access appropriate community services [15,16]. While formal psychological support delivered individually or in groups can be of benefit [17–19] these interventions are typically delivered by specialists which further limits accessibility. Understanding the subjective experiences of family members is essential for developing novel support methods that are less reliant on specialist staff.

The 'Life Thread' model provides a potential basis for such support following brain injury [20] and is grounded in narrative theory, constructionism, and the sociology of biographical disruption [21–25]. The Life Thread model emphasises that individuals maintain a sense of self and coherence through the "threads" of their life stories which are created with others. The thread metaphor relates to the life narratives which are broken or disrupted (frayed). When disrupted, individuals need to actively work to re-create a sense of coherence. The understanding gained is closely aligned with narrative identity and identity reconstruction in mental health [26–29] and the possibility of post-traumatic growth [30]. The Life Thread model also resonates with the Family Tasks Model [31] which places meaning making at the centre of the process of adaptation for families after Acquired Brain Injury (ABI). The model has informed an arts and health intervention to support identity changes for stroke survivors [32]. It has also been overlaid with narrative dimensions in a meta-synthesis of research into TBI family experiences indicating potential clinical applicability with families affected by TBI [33]. However, its application as a tool to support family adaptation after TBI has not been empirically investigated.

Co-developed with families affected by TBI, the Life Threads Approach (LTA) gives the Life Thread model physical form using materials people can use at home. As both a symbolic and practical tool, it supports exploration of continuity, disruption and transformation in the aftermath of TBI, encouraging reflection and conversation. Consistent with creative arts approaches [34] the LTA enables individuals to physically manipulate representations of their life narratives. We sought to explore if this

embodied interaction could help them gain insight into unconscious or less articulated aspects of their past, present, and future identity.

Accordingly, this study aimed to understand the clinical potential of storytelling through the LTA and gather the information required to plan a feasibility randomised control trial. The primary objective of this qualitative pre-feasibility study was to explore if family members' find storytelling through the LTA useful as a strategy to support their individual subjective wellbeing and adjustment post-TBI (See Table 1).

## Methodology

This was an in-depth qualitative study situated within an interpretivist paradigm with a relativist ontology and constructivist epistemology consistent with the narrative orientation of the LTA [35,36]. This approach drew on our strengths and values as qualitative researchers reflected in our prior work to understand and improve the experiences of family members impacted by brain injury and to do so in an inclusive way.

### Ethics

The study received favourable opinion from Nottingham 1 Research Ethics Committee (23/EM/0185) and HRA approval on 4 September 2023. Registration with ISRCTN (Trial ID: ISRCTN17392794) followed and the protocol published [37]. Key ethical considerations included participant well-being and the injured person's awareness of their relative's involvement. All participants were signposted to organisations that could offer support after emotionally sensitive interviews and focus groups. In addition, a clinical psychologist was made available to participants who experienced more severe distress following study activities. One participant requested this intervention. All study procedures were made transparent to survivors; however, their explicit consent for family member involvement was not requested. Nine non-substantial and one substantial amendments were made, initially these were technical amendments, and later to enhance recruitment through new sites, revised materials, and an extended study period.

### Patient and Public Involvement and Engagement (PPIE)

Study PPIE is reported in line with the Guidance for Reporting Involvement of Patients and the Public (GRIPP2 Short Form) [38] (Table 2). Two PPIE representatives were co-investigators, and four additional family members formed an advisory group. Half the group (n = 3) were female and half from minoritised ethnic backgrounds. Co-author AN led the PPIE workstream, coordinating support and administration. The advisory group met quarterly together, and then with the wider research team four times through the project life cycle. The group contributed to the development of the LTA, reviewed participant-facing materials, supported recruitment, and provided feedback during analysis and dissemination to align

**Table 1. Study objectives.**

| No. | Objective |
|---|---|
| I | To explore if family members find storytelling through the LTA useful as a strategy to support their individual subjective wellbeing and adjustment post-TBI. |
| II | Assess uncertainties in relation to the clinical application of the 'Life Threads' approach including: acceptability; adherence; and level of facilitation required. |
| III | Identify appropriate methods for a feasibility study including: representative recruitment; choice of primary outcomes; mode of delivery; and comparator arm(s). |
| IV | Understand how family members use the 'Life Threads' approach to understand the impact of TBI on themselves and their families. |
| V | Explore if the four domains of subjective experience post-TBI (Displacing and Anchoring; Rupturing and Stabilising; Isolating and Connecting; Harming and Healing) are representative of family member experiences. |

**Table 2. GRIPP2 short form.**

| Section and Topic | Item | Life Threads-TBI Methods |
|---|---|---|
| 1: Aim | Report the aim of Patient and Public Involvement (PPI) in the study | To ensure the study was grounded in the lived experience of families after TBI and to enhance the LTA prior to recruitment and data collection. |
| 2: Methods | Provide a clear description of the methods used for PPI in the study | Two family members were co-investigators, with four additional members forming an advisory group. Families were consulted prior to funding and involved throughout the research process. |
| 3: Study Results | Outcomes—Report the results of PPI in the study, including both positive and negative outcomes | PPIE input significantly improved the LTA, particularly in presentation, support materials, and recruitment (including a co-produced recruitment video: https://tinyurl.com/4stfbn5c). Members also contributed to data interpretation, drawing on lived experience. |
| 4: Discussion and Conclusions | Outcomes—Comment on the extent to which PPI influenced the study overall. Describe positive and negative effects | PPIE members felt heard and valued, highlighting a sense of inclusion and mutual respect. The group evolved into a supportive, collaborative team. Challenges included temporary withdrawals from the advisory group due to caring responsibilities and other critical life events, reflecting the real-life complexity of life after TBI for family members. |
| 5: Reflections/ Critical Perspective | Comment critically on the study, reflecting on the things that went well and those that did not, so others can learn from this experience | Involving carers brought both deep insight and logistical challenges. PPIE training and onboarding requirements (including the need for references) also created a barrier to effective engagement. |

the findings with their lived experience. The PPIE evaluation showed group members felt that the experience of working together provided them with a sense of purpose and identity outside of their roles as carers for their loved ones.

## Setting

The study was initially conducted in community settings across the East and West Midlands; however, recruitment was later expanded to all regions in England. While some research team members had prior relationships with recruiting sites, none had worked with study participants.

## Sample

We had planned to use maximum variation sampling, stratified to reflect regional diversity and family member characteristics; however, despite extensive engagement with gatekeepers and revising our recruitment materials, we finally applied simple non-random, purposive sampling using the criteria listed in Table 3. We used an inclusive definition of family as 'the family is who they say they are' [39 p. 60] so that any person who identified as a member of the injured person's family, including a close friend, was eligible to participate and more than one person per family could take part. There were no restrictions based on injury type, family role or the survivor's level of impairment. Only those family members with mental health issues of a nature or severity that jeopardised safe engagement in the study tasks were not eligible to participate. These risks were explored during the meeting prior to consent.

Sample size was determined by the principles of data sufficiency' by Dey [48], in contrast to data saturation which is not in keeping with our analytical framework [47]. From the outset, a sample of 20 participants was deemed sufficient to address the aims and objectives of this study.

**Table 3. Study inclusion and exclusion criteria.**

| Family member | Inclusion Criteria |
|---|---|
| Identifies as a family member or close friend of a person with: | Any severity traumatic brain injury, sustained at least two years prior, age at injury 18 years or older. *(If approached by a family member of a person between the age of 16–18 at the time of injury who was treated within adult services, we will consider this family member eligible to take part).* |
| The family member must be: | • Known to the injured person before injury.<br>• Age 16 years or above.<br>• Able to give informed consent.<br>• Residing within England<br>• Have access to a smart phone, tablet or computer that can access the internet.<br>• Willing to participate in a group.<br>• Fluent in English. |
| | **Exclusion Criteria** |
| | Those with mental health issues of a nature or severity that jeopardise safe engagement in the study tasks. |

## Recruitment

Participants were sought through National Health Service and third sector settings. Study adverts were posted on social media and reposted by key organisations including United Kingdom Brain Injury Forum (UKABIF), Headway UK and regional Headway groups and branches, the British Association of Brain Injury Case Managers and Anchor Point, a national initiative driving change for families after ABI (anchorpointabi.org). Study adverts were also sent to the mailing lists of these organisations. On receipt of an 'expression of interest' each person was sent a detailed participant information sheet and consent form. A follow-up meeting was scheduled to confirm eligibility, answer any questions and take verbal confirmation of each consent item in the informed consent form. The consent form was then sent to participants via DocuSign for electronic signature. On completion of the study participants were given a £25 online shopping voucher and reimbursed for out-of-pocket expenses.

## Dataset generation (Data collection)

Following informed electronic consent, participants completed a demographic questionnaire about themselves and their injured relative. Participants then joined one of seven online focus groups, averaging 55 minutes (range: 39–81 minutes), facilitated by CJW and a second researcher (FG or CEH). The topic guide, developed with input from the PPIE advisory group, introduced the LTA and provided an opportunity to meet others, share experiences and answer questions. The LTA, consisting of woollen yarn, blank labels, a guide (encouraging participants to use the LTA in their own way), and pictures of the LTA representing life stories was mailed to participants in a letter-size box.

Approximately one month later, participants took part in an unstructured in-person interview with CJW to share their TBI story using the LTA. Follow-up questions were based solely on topics raised by the participants and averaged 74 minutes (range: 45–142). Interviews were mostly conducted in participants' homes, with four held in a confidential room in an educational setting. Written consent was requested to photograph the threads and labels; however, one participant declined due to concerns about misinterpretation.

The final focus group facilitated online by CJW and one other researcher (FG, CEH and NY) averaged 74 minutes (range: 65–81 minutes). Topics discussed included what worked well, what did not, what improvements could be made, if there were any benefits and if this could be used to help others, see published protocol for further details and images of the LTA [37].

## Analysis

Interview and focus group data were analysed using Braun and Clarke's [40] reflexive Thematic Analysis (TA) whereby the researcher is seen as a tool within the analytical process. As such a positionality statement is provided below.

## Positionality statement

CJW led the analytical process and is a White female Professor of Nursing and experienced qualitative researcher. CEH reviewed all transcripts and engaged in critical dialogue with CJW throughout. CEH is White female academic with an occupational therapy and psychology background and expert in qualitative research. All findings were discussed with senior members of the research team, AN, a White female Associate Professor of psychology with lived experience of familial brain injury, and FG a White male Associate Professor in clinical psychology with a background in neuropsychological rehabilitation clinical practice and research. Early themes were shared with the wider team and PPIE representatives. The wider team reflected an interdisciplinary perspective across the acute-community brain injury pathway, all of whom share a passion to improve the lives of families after brain injury that will have influenced our interpretation of the data. As people with clinical, research and personal experience of brain injury, the core research team have developed a commitment to improve understanding of the needs of family members impacted by brain injury, and to promote ways in which family members can be included in rehabilitation and receive support in their own right. CJW and AN are co-chairs of Anchor Point, JCW, NY, MH, SR and FG are also members.

## Analytical process

Audio files were transcribed by an approved third-party and checked by CJW for accuracy and anonymisation. While the analysis focused on the study objectives, an inductive approach allowed openness to participants' priorities. The six stages of reflexive TA are summarised in Fig 1 and detailed in the Table 4.

## Rigour

Analytical rigour was grounded in a 'Big Q' approach commensurate with a qualitative paradigm [41]. We used Braun and Clarke's 15-point checklist for reflexive TA [40], adhered to reflexive TA reporting guidelines [42] and were guided by Tracy's 'Big Tent' criteria [43]. See detailed responses in the Supplementary files S1–S3 Tables.

## Findings

The recruitment period for this study spanned February – June 2024. In total 59 expressions of interest were received. Of these, 9 did not follow-up after receiving the information sheet, 16 were ineligible (10 due to injury type, 3 due to time post-injury, 3 for other reasons). Concerns were raised with the sponsor about expression of interest with patterns/phrasing not typical of genuine correspondence. Following closer scrutiny of these emails 13 were deemed to be phishing attempts. One family member withdrew after consenting.

Ultimately, 20 family members of 17 individuals with TBI enrolled; three later withdrew after the first focus group, leaving 17 who completed the LTA and second focus group. Informal contact with the three participants who withdrew suggests it was not the right time to explore the impact of TBI on themselves in this way. This sample size was within the parameters deemed sufficient to address the research aim [37,44].

## Sample demographics and injury type

Of the twenty participants, all spoke English as their first language and 19/20 were White. Only one participant was recruited via an NHS site. Family members were predominantly female (85%, n = 17) and injured persons mostly male (65%, n = 14). Half (n = 10) of the family members were aged 25–54 and half (n = 10) aged 55–84, 71% (n = 12) of

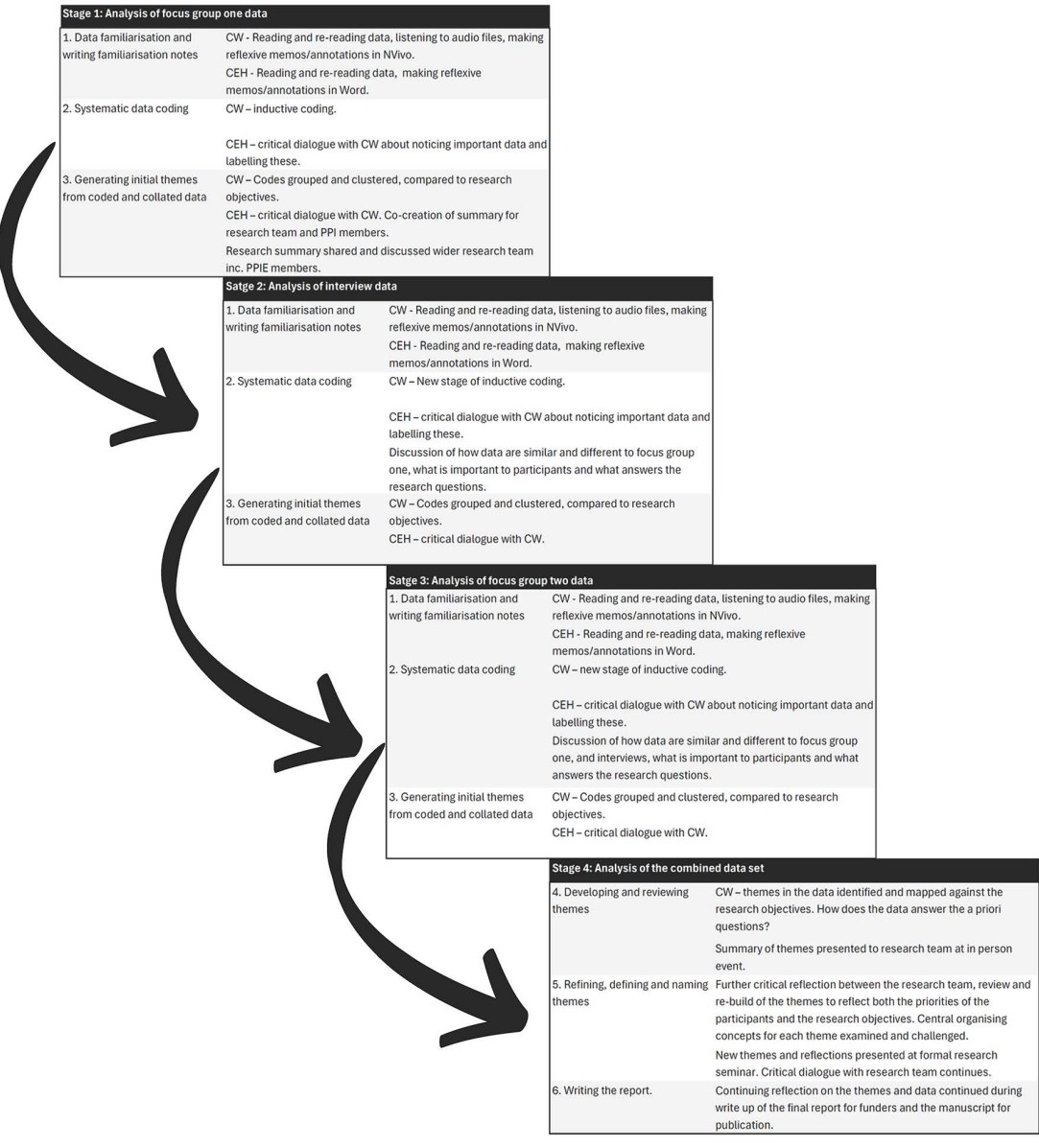

**Fig 1. Process of reflexive TA.** Illustration of engagement with the six stages of reflexive TA.

injured relatives were aged 25–54 years and 30% (n = 5) were aged 55–84. Half of the sample (n = 10) were spouses or partners of the injured person, n = 5 were parents, n = 3 were their children and n = 2 were siblings. The severity of injury was determined by family members who identified these as mild (n = 1), moderate (n = 2), severe (n = 13) or unknown (n = 1). Time since injury spanned two – 30 years with almost half (n = 9) 2–5 years post injury. Only one injured survivor lived at home with full time care, and two resided in a long-term care facility, eight lived independently from the family member and five lived at home with the family member. The demographics of all participants are summarised in Tables 5 and 6.

**Table 4. Analytical activities enacting reflexive TA.**

| Step | Analytic task | Analytic activities |
|------|---------------|---------------------|
| 1. | Data familiarisation and writing familiarisation notes | CJW conducted all interviews and focus groups, listened to the all the audio recordings and read/re-read transcripts. Reflexive annotations and memos were recorded in NVivo.<br>CEH read/re-read transcripts and made reflections the margins. |
| 2. | Systematic data coding | Formal coding was led by CJW, who used this process to identify important, interesting, and meaningful data that both answered the research questions but also felt important to the participants and their own experiences. This meant that coding was both semantic (descriptive), and latent (interpretive). Coding did not produce a fixed code book. Instead, aligned to Braun and Clarke's conceptualisation, coding was an 'organic and evolving process of noticing' tagging data and then building codes from which themes were later developed.<br>This process of noticing, tagging and coding was shared with CEH in regular meetings throughout the analysis process. |
| 3. | Generating initial themes from coded and collated data | Data from focus group one, individual interviews and focus group two were treated as separate data sets. With the process of inductive coding commencing anew, rather than forcing different data to fit an a priori coding scheme. This allowed us to notice new meanings relevant to the context in which the data were collected. CJW and CEH discussed 'candidate themes' through the clustering of codes, relationships and hierarchies. These initial themes from each data set were critically discussed with AN and FG. |
| 4. | Developing and reviewing themes | In reflexive TA 'a theme captures a shared meaning, united by a central organising concept' (Braun and Clark, 2022, p.77). Therefore, in stage four CJW explored the initial themes and worked to bring these together in a way that would answer the research questions and retain a commitment to the priorities of the participants. This involved the exploration of shared/similar meanings across the different data sets. We aimed to reach an understanding that would 'dig below the surface meaning'. These themes were taken to a co-investigator meeting that included members of the PPIE advisory group and wider research team. At this meeting data under pinning interpretation and answers to the research question were critically discussed. |
| 5. | Refining, defining and naming themes | Findings from the analysis were organised within 'overarching themes' to answer the research objectives, under which themes and sub-themes were confirmed. This final analytical step was discussed with all members of the research team and PPIE group. |
| 6. | Writing the report. | Stage six was a final opportunity to refine the analysis and shape the 'story' to be told from this research. This was as much an interpretive and creative step as the others and influenced by our positionality, understanding of the field and the audience for whom this story is told. |

## Description of Life Threads engagement

Fifteen of seventeen participants engaged broadly with the LTA as suggested and two engaged differently. Half used over ten labels to help tell their story and only one did not use these at all. Labels were used in a variety of ways from single words to full sentences relating to adaptation, growth, hope, people, support, roles, emotions, feelings, relationships, change, difficulties, positive and negative evaluations, milestones, critical events, affirmations, timestamps, reflections and children. Six family members chose to add something extra to the LTA including: a poem, a song, photographs, a spreadsheet. One participant chose to make the size of the labels relevant to their story, casting negative thoughts onto smaller labels and positives ones onto larger ones. Thirteen participants said that their interactions were staged or ongoing, returning to the materials in a physical or metaphorical sense during the four weeks.

Eleven participants reported tangible benefit from engaging with the LTA, while two described qualified benefits, one noting short-term impact only, another attributing benefit to the process rather than the LTA. Two saw no benefit, citing prior therapy, and for two the impact was unclear. When asked if the LTA enabled storytelling in a way traditional methods could not, thirteen participants said it allowed them to tell their story differently. A descriptive summary is provided in Table 7.

## Reflexive themes

Reflexive TA enabled exploration of participants' priorities alongside the study objectives. The resulting themes, summarised in Table 8, are rich and complex, illustrating both the context (Theme 1) and changing needs of families after TBI

**Table 5. Sample demographics.**

| | | Focus group 1 | | LTA & Focus group 2 | |
| --- | --- | --- | --- | --- | --- |
| Variable | Category | Family member (n = 20) | Injured person (n = 17) | Family member (n = 17) | Injured person (n = 14) |
| First language | English | 20 | 15* | 17 | 14 |
| Ethnicity | White | 19 | 17 | 16 | 14 |
| | Mixed | 1 | 0 | 1 | 0 |
| Gender | Male | 3 | 13 | 3 | 11 |
| | Female | 17 | 4 | 14 | 3 |
| Age | 25-34 | 2 | 4 | 1 | 2 |
| | 35-44 | 7 | 5 | 7 | 5 |
| | 45-54 | 1 | 3 | 1 | 3 |
| | 55-64 | 4 | 2 | 3 | 2 |
| | 65-74 | 5 | 1 | 5 | 1 |
| | 75-84 | 1 | 2 | 0 | 1 |
| Relationship to injured person | Spouse | 10 | | 8 | |
| | Parent | 5 | | 4 | |
| | Child | 3 | | 3 | |
| | Sibling | 2 | | 2 | |
| Sexual orientation | Heterosexual/ Straight | 20 | 16 | 17 | 13 |
| | Lesbian/Gay | 0 | 1 | 0 | 1 |
| Marital status | Married | 18 | 11 | 15 | 9 |
| | Living with partner | 2 | 2 | 2 | 1 |
| | Divorced or separated | 0 | 1 | 0 | 1 |
| | Single | 0 | 3 | 0 | 3 |
| Disability | No | 20 | | 17 | |
| Religion | No religion | 7 | 9 | 7 | 8 |
| | Christian | 13 | 7 | 10 | 5 |
| | Prefer not to say/ unknown | 0 | 1 | 0 | 1 |
| | | *n = 2 missing data | | | |

**Table 6. Family member reported injury characteristics of survivors.**

| Characteristic | Category | Focus group 1 | LTA & Focus group 2 |
| --- | --- | --- | --- |
| Severity of Injury | Mild | 1 | 1 |
| | Moderate | 2 | 2 |
| | Severe | 13 | 10 |
| | Unknown | 1 | 1 |
| Time since injury | 2 - 5 years | 9 | 7 |
| | 6 - 8 Years | 2 | 1 |
| | 9 - 20 years | 4 | 4 |
| | 21-30 years | 2 | 2 |
| Living arrangements | Lives at home requires full time care | 1 | 1 |
| | Long-term care facility | 1 | 1 |
| | Long-term care facility (now deceased) | 1 | 1 |
| | Lives independently from participant | 5 | 4 |
| | Lives at home with participant | 8 | 7 |

**Table 7. Descriptive summary of LTA engagement (n = 17).**

| Engagement | Descriptive Summary | |
|---|---|---|
| Did they work with the LTA as suggested? | Yes | 15 |
| | No* (*Still engaged but not as suggested) | 2 |
| No. of labels used. | 0 | 1 |
| | < 5 | 1 |
| | 5 - 10 | 5 |
| | >10 | 10 |
| What went on the labels? | Adaptation, growth, hope, people, support, roles, emotions, feelings, relationships, change, difficulties, positives and negatives, milestones, critical events, affirmations, timestamps, reflections, children | |
| Did they add anything? | Yes | 6 |
| | No | 11 |
| | Examples: Photographs, poem, size of the label; song, shapes, spreadsheet | |
| Did they interact with the LTA multiple times? | Yes | 13 |
| | No | 1 |
| | Unclear | 3 |
| Did the participant find the LTA beneficial? | Yes | 11 |
| | Qualified yes* (*yes but not long term and yes the process but not the LTA specifically) | 2 |
| | No | 2 |
| | Unclear | 2 |
| Did the LTA 'add value' to the re-telling of their story? | Yes | 13 |
| | No | 3 |
| | Unclear | 1 |

**Table 8. Reflexive themes and associated sub-themes.**

| Theme 1 | Theme 2 | Theme 3 | Theme 4 |
|---|---|---|---|
| **Scaling Cliffs with Broken Wings** | **An entanglement of wellbeing** | **Hear me, See me: The Power of Story** | **Creating the conditions for stories to be told** |
| *Research objective V* | *Research objective V* | *Research objectives I, IV, V* | *Research objectives II, III* |
| **Broken:** "sad beyond sad" (Res11) | **A relentless commitment to recovery:** "I have got to step up" (Res13) | **Subconscious and conscious processing:** "I'd got lots of things that I was processing in my head" (Res5) | **Safe spaces:** "I'm not a creative person at all" (Res2) |
| **The second insult:** "they don't understand what brain damage is" (Res4) | **A loss of self:** "I know I'm not the person I was before the brain injury" (Res12) | **Noticing, naming and symbolism:** "this is us" (Res8) | **Bearing witness:** "no one's ever asked me how I was" (Res6) |
| **Fight or fall:** "we're going to hang on in there and fight" (Res20) | **Attending to self:** "the work is never finished" (Res9) | **Steps towards healing:** "I've got to sort out a narrative" (Res4) | **When words are not enough:** "we've been through all of this trauma, and you're giving me some string?" (Res8) |

(Theme 2). Through this process, many realised they had never told 'their story' before and the LTA provided both the permission and the tools to do so (Theme 3). The final theme explores the conditions in which benefits such as voice, agency, and sense-making can emerge (Theme 4). Themes are presented below with relevant de-identified extracts of

raw verbatim data, where quotes are shortened for brevity […] is used. Images are purposely presented without the corresponding respondent codes to enhance their anonymity.

**Theme 1: Scaling cliffs with broken wings**

This first theme is an orientation to the people and the problem to set the scene for why families need support following TBI. The sub-theme **'Broken'** captures the toll of TBI on families. **'The second insult'** explores how trauma is compounded by the actions of those closest to them. Finally, **'Fight or fall'** highlights the varied consequences for families from a determination to drive change, to exhaustion and a sense of defeat.

**Broken: "sad beyond sad" (Res 11).** Seeing a loved one cheat death only to have to face decisions about removing life sustaining treatment or a prognosis that robs you of the person you once knew is devastating.

> *Res5 FG1 "They actually told me if he survived the night- I had to call his parents in and our children in at two o'clock in the morning. And they said if he survived [distressed]. That he'd never be normal again [distressed]"*

These wounds are deep, and family members often expressed a sense of traumatization, self-identified post-traumatic stress disorder, depression, and anxiety. The articulation of loss and the complex emotions that accompany grief were a hallmark of many stories. Recognising loss and making way for grief were not easy to accept nor share with others.

> *Res11 Int: "One of the therapists said that, "You're grieving who you had." I can't even think of that because he's still here you know but she said it is probably what you're doing".*
>
> *Res9 Int: "So, the big one is obviously grief, and I miss him, but I've still got him, he's just different, yes. He had big dreams and they've all gone. […] She didn't want to see the one that I had written about grief and that I miss <him, Son>"*

The consequences of TBI endured for many years with needs vacillating over time, new traumas to be faced, and buried emotions to surface.

> *Res8 Int: "we had to have all the trees cut down in my garden which was obviously horrific for the kids knowing why we had to have them all taken down because he was trying to hang himself from them"*
>
> *Res2 Int: [30 years post-injury] "But yes I would say it's only been the last two or three years that I've actually felt really content with my life and how it is"*

When normal resources failed, family members described their own, and their family's point of breakdown searching for answers that never came. For some, formal therapy was a means to find these answers and understand the true impact of the past on the present.

> *Res1 Int: "I needed this therapy 10 years prior. […] I thought I was going into therapy just to talk about general stuff and every single session ended up being my mum and how I felt about things, for six months. There wasn't a session where we didn't talk about that"*

Without this, feelings hidden away, buried deep, manifest as fragility, anger and self-destruction.

**The second insult: "they don't understand what brain damage is" (Res4).** The second insult stemmed from actions and behaviours from outsiders looking in. Family members described lack of understanding from society, family and friends that left them feeling judged, unseen, dismissed and unvalued. Unlike the first trauma, this secondary trauma

was often preventable and thus even more difficult to make sense of. The perceived lack of insight and understanding from the outside world created a void between those with lived experience and those without. The experience of living with a family member with a brain injury felt too difficult to comprehend, too long to sustain interest, and too complicated for simple gestures of support.

> *Res4 FG2: "People with cancer, it's, "Oh gosh, cancer, of course you're going to help your father," but brain damage is- Everyone knows what a brain is but they don't understand what brain damage is. And even though it's obvious, you have to spoon feed it to people"*

> *Res14 Int: "on really bad days I just feel like I'm completely trapped and married to a man I have no idea who he is. But no one on the outside can really see that"*

The lack of societal understanding was the reason that many family members wanted to take part in this study and was understood as a contributing factor for the lack of available support. Family members were left feeling judged or abandoned. Tailored support for children was another void where need was ignored.

> *Res8 Int: "His parents were very much, "You should be over your brain injury in the first couple of years," and they expected him to take over a family business which he just wasn't able to do"*

> *Res3 Int: "Yes I just felt like, and literally the minute he came out of the hospital the doors were shut and that was it we were literally left to it"*

> *Res1 Int: "They never said she was going to be a different person or she wouldn't be able to understand if we were feeling sad. Nobody, at any point, talked to us about that"*

Many family members described painful encounters with professionals, feeling labelled as the 'bad family' (Res20), treated like 'the enemy' (Res9), or excluded from decision making, dismissed or punished.

> *Res20 Int: "But the frustrating thing is the ICB\* it's like trying to attack a fog. They have control but you don't know who they are and I think that's not good"*

> [*An ICB is an Integrated Care Board which commissions health services in England]

> *Res15 FG1: "and I'll think to myself, why did nobody tell me it was going to be like this?"*

These stories revealed stark power imbalances, where families felt dismissed, despite being relied on to provide care. The lack of power, influence and ability to advocate for the needs of their relative is captured in this quote from Res13.

> *Res13 Int: "But someone with a job role, doing exactly the same thing, would be met instantly with a response. Like, so I was trying to get a response for like, four weeks, and they got a response the next day. And when you continuously see that happening… [becomes upset]"*

Treatment by professionals in this way left an imprint on how that family member perceived themselves, devalued their role as a carer and further reduced their sense of agency and power.

**Fight or fall: "we're going to hang on in there and fight" (Res20).** Hope plays a complex role in participants' experiences, fuelling a desire to fight the odds and the system, yet for some, having nothing left at the end.

> *Res11 Int: "On occasions I had thoughts it would be easier not to wake up"*

*Res20 Int: "You could almost say it's broken \<him\> and he's a strong guy and I think that's disgusting"*

Within the context of brain injury, family members often talked about the lack of certainty which fuelled hope, resolve and a desire to keep fighting. Over time, this fight wasn't always possible to sustain, and some reflected on the notion of false hope. Letting go of hope allowed some to come to terms with their circumstances and move forward. Hoping for the future felt fragile, distant, and tinged with sadness evolving in response to their loved one's recovery and their changing circumstances.

*Res9 Int: "The consultant very gently said, "I'm all for dreams but that is not going to happen, we need to have a bit of realism here." At the time, I just thought, that's just silly, but he was right, you know…, which is hard to accept really […] So, we just have to let go of the stuff that we've had to let go and move forward…or just tread water really... [pause]"*

*Res8 Int: "if you'd have asked me to do this even three years ago you wouldn't have had any of the new hope on it I don't think. I don't think we saw much of a future. Whether we would have done working through this or not but I do think it takes a long time sometimes"*

Nested somewhere between having the energy and appetite to fight and the utter exhaustion and desire to give up was also a compulsion to use their experiences to help others.

**Theme 2: An entanglement of wellbeing**

The second theme helps to understand why family interventions, such as the LTA, may fail to work as intended by examining **'a relentless commitment to recovery'**, **'a loss of self'** and the importance of **'attending to self'**.

**A relentless commitment to recovery: "I have got to step up" (Res13).** Family members described their dedication to the well-being of the entire family system. Family members' initial focus is securing the best possible outcome for the injured person, actively engaging in rehabilitation and stepping in to fill gaps in service provision. This involvement was not a choice but an unquestioned act of love and responsibility.

*Res5 FG2: "And I did say to [interviewer], as long as me, \<husband\> and my boys are all right, everybody else is a bonus. But they're not my priority anymore. And it makes you do that, it makes you hanker down a little bit..."*

*Res13 Int: "I have put on there as well I have to care for him but it's my job, you know. I just feel, like I said to you, I felt like it has got to be me"*

*Res18 Int: "I think I was always of the view that you had children to get them to the point where they were going to fly the nest and if you'd done your job properly they could manage. I think I'm probably doing the same again for \<Son\> whether you know, I don't expect him to fly the nest very far but so I'm doing that, I'm trying to make him as independent and as yes, independent and as fulfilled as he can be"*

Family members saw themselves as 'the bottom line' positioning themselves where they could be the strength that others needed. This responsibility led to role change and new responsibilities. Family members wanted to learn, understand, be involved, and became experts in the needs of the injured person.

*Res11 Int: "We became involved in every part of therapy we could, especially when the hospital didn't have the resources to do it"*

Once the injured person had reached a point of stability, attention could then turn to the needs of others including children and aging parents.

*Res12 Int: "But I had a two-year-old. No communication, which her communication is still pretty poor, and she's seen everything. She's heard everything, she's watched everything, she's seen what I've seen. So my focus is on her now"*

This immersion and dedication to the injured person and the wider family system means there isn't always room for the family member to craft their own story or meet their own needs.

**A loss of self: "I know I'm not the person I was before the brain injury" (Res12).** Life cast in the shadow of brain injury felt very different.

*Res9 Int: "it's just a different life, it's just very different"*

Immersion in brain injury and their loved one's recovery left many family members feeling they had changed. These differences were in part due to the self-sacrifice and unquestioned commitment and balancing the needs of other people in the family system leaving little to no room for their own. It often took time to notice the effects of this commitment or be willing to accept help.

*Interviewer: "is there anything in here about you, and hope for yourself?" Res12: "No. There's none […] There was no hope for myself, and that sort of like went for myself there really, it's all been for my husband and children"*

This commitment comes at a cost with some family members not recognising themselves or their relationships over time.

*Res5 Int: "I'll never be the person again. So I just can't bring myself to have it cut off [talking about her hair] because I know that the person I look at in the mirror's not going to be the same"*

*Res16 FG1: "I think for the first four years after the accident, she and I were incredibly close, I think because she needed me. But then the last couple of years, it just seems like all we do is argue"*

While some changes felt painful, others reflected personal growth, such as increased strength and compassion for others.

**Attending to self: "the work is never finished" (Res9).** The ability of the family members to attend to their own wellbeing or accept care from others was often limited. Self-care often felt like another unwelcome task, something that *should* be done. Therefore, making room to complete the LTA was hard for many with competing priorities.

*Res13 Int: "But again, it's very difficult focusing on yourself and doing something for yourself because you have to put a lot of effort to make time for that, and then you feel bad because- Well not necessarily feel bad but you feel like you're just juggling everything and making time for more stuff, which we should maybe be putting less on my plate maybe than more, and focusing on myself feels like a job, if that makes sense?"*

Recognising the existential shift, as a threat to their own wellbeing, was crucial in influencing their willingness to prioritise their needs. When participants saw their well-being as directly influencing the well-being of others, the drive for self-preservation became stronger. Yet, the guilt of taking time for themselves made self-care difficult.

*Res3 Int: "I will just look at some that might be more, my needs aren't important, that's how I felt at the time. But now I am saying I have got to think of myself, which I have because I am no good to him if I don't think of myself"*

The LTA required family members to make some time so they could see their own story and connect to their own sense of self and this was seen as beneficial.

*Res5 FG2: "It was the first time in four and a half years that actually, it was for me. It was about me"*

By connecting to their story in this way family members recognised the interdependent and relational cycle of wellbeing where family members affected one another.

**Theme 3: Hear me, see me: The power of story**

The third theme examines the value of storytelling through **'Subconscious and conscious processing'**, **'noticing, naming and symbolism'** and **'steps towards understanding'** made possible through the LTA.

**Subconscious and conscious processing: "I'd got lots of things that I was processing in my head" (Res5).** For many family members the process of reflection started in the first focus group and then more specifically as they received the LTA materials. This gentle invitation to start the process of reflection and consider the story of TBI from their own perspective gave people the opportunity to pause. For some, subconscious initiation was followed by a more conscious act of processing where family members purposefully and actively thought about how TBI had affected their own lives. Thoughts were manifesting at a subconscious level and then later, when working with the LTA, these thoughts could be processed more deeply. Several family members talked about the need for time, and felt it was important to do so in a way that would not be too painful.

> *Res6 FG2: "I needed time to almost think about it and process it"*

> *Res9 Int: "I tried to do it when I was in a positive frame of mind rather than a negative"*

The ability to engage, step away and re-engage at another time was key for many. This 'staged' interaction with the LTA felt like a way of protecting themselves from the risk of delving too far too fast, or when returning to their threads with others meant participants could step into a deeper form of reflection but with support. The LTA itself was described by one participant as '*a starter kit to start a conversation*' (Res6) emphasising the importance of facilitation and that sense making was a process not an end point.

> *Res9 FG2: "Then I put it back in the box and put it in the drawer, and you know. Yes. […] we've done that emotion lets, let's put it back in the drawer and crack on with you know, whatever else we have to do"*

> *Res5 FG2: "maybe for future ones is just perhaps take on almost like, 'Previous people who have been part of the study have found it useful to do this, walk away and then come back to it'"*

Through both subconscious and conscious processing family members revisited memories with the opportunity for new insights and growing appreciation for how these events and experiences had shaped their lives and their future. These experiences show the need for the LTA to be part of a process where family members can engage, pause and return.

**Noticing, naming, and symbolism: "this is us" (Res8).** Following the commitment to engage with the LTA came the act of noticing what was central to their story, after which participants could name these facets and give them presence in their story. All but one person wrote on the cards provided. These storied elements were cues to dig deeper and helped family members to find connections and meaning (see examples of LTA noticing and naming).

**Examples of LTA noticing and naming:**

> *"and then everything changed... So at the point of <Son>'s injury the first things are fear, panic, ignorance, pain and in time the hardest thing is you realise that you can't put it right… [very upset]… […] so there was a feeling of loss for his future and ours and concern about supporting our daughter and the elderly family members we had left." (Fig 2)*

> *"Let's see what I put on here, I've put on here that it's been six and a half years, remarkable recovery, heartbreaking situation. I put some backgrounds about our family, stories from the past, I've picked out the little main points.*

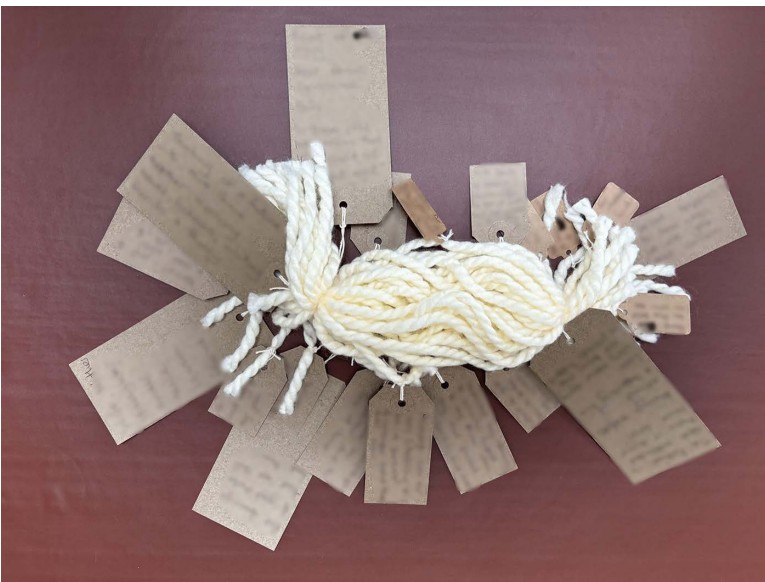

**Fig 2. LTA noticing and naming (image a).**

*I've put about preventative focus, being unhappy and depressed so got him home rather than six months in another rehab."* (Fig 3)

*"I kind of thought about major things and it's a bit loose, you know, but kind of bigger things and how they were connected to other things in what's happened, you know. And, for example, you know, one of the big labels just might say, "The Accident," and what happened and the story of what happened and stuff like that"* (Fig 4)

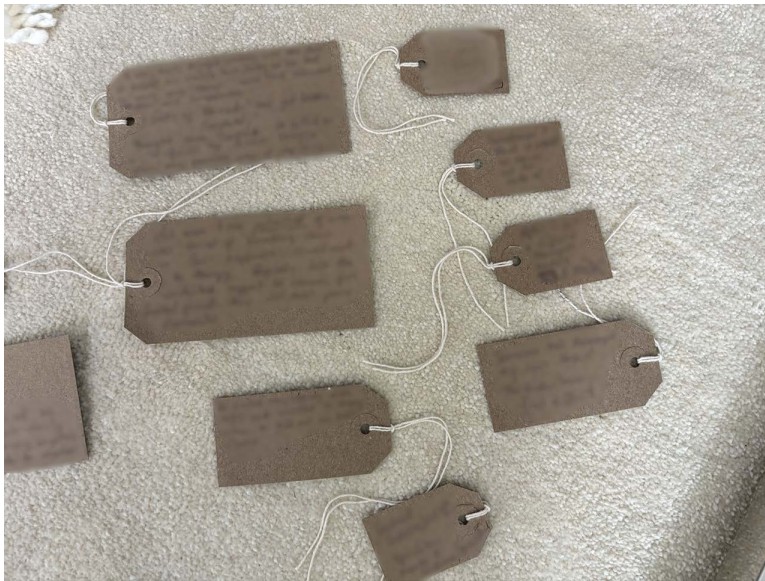

**Fig 3. LTA noticing and naming (image b).**

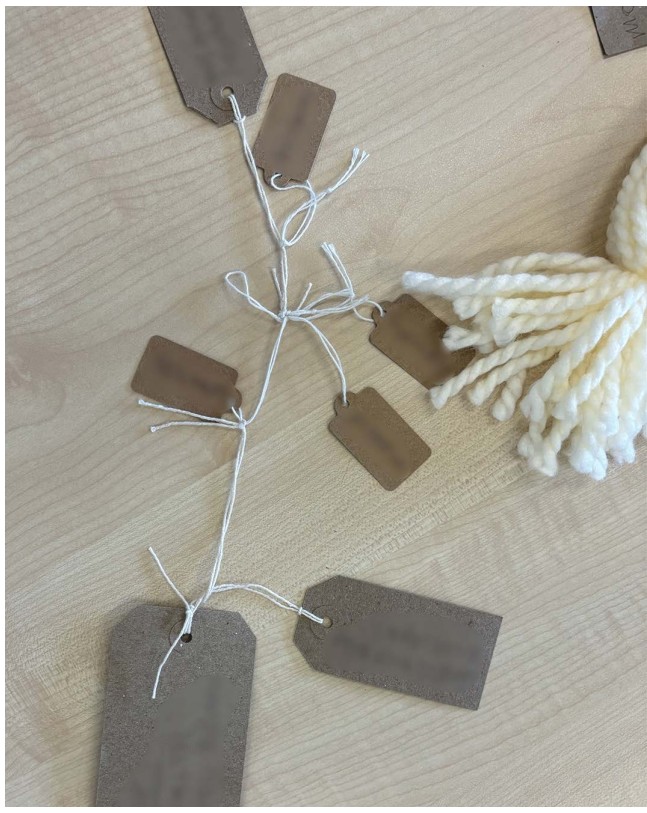

**Fig 4. LTA noticing and naming (image c).**

Drawing attention to certain parts of themselves and their story was interpreted as agency and voice and for a few this part of the LTA was considered more useful than the threads themselves. For most people the act of writing on the labels made the meaning of these words clearer and often more profound.

Family members wrote down their feelings, emotions, roles and relationships, timestamps, critical events, people, concerns, fears and questions. Participants noted new feelings of empathy, growth, wrote about hope and their aspirations for the future. Some family members deconstructed and reconstructed their threads mapping out their story with detail and meaning (see examples of deconstruction and reconstruction).

**Examples of deconstruction and reconstruction:**

"*then all of a sudden it transformed into, this is threads. This is us. This is the brain injury. This is how this connects. At every point where they made a connection and had a memory or needed a feeling put in, sparked more conversation*" (Fig 5)

"*It could have just been one long one, but I needed… This hope was important, and it was important connecting, I know you're there to year three. So that's where the shape came through really*" (Fig 6)

"*I've split this into five scenarios or time slots […] So according to the wool because it's intact so it's how close we feel to others and their stories. […] So I had the blue of helplessness and I then felt the displacing and loss of stability*" (Fig 7)

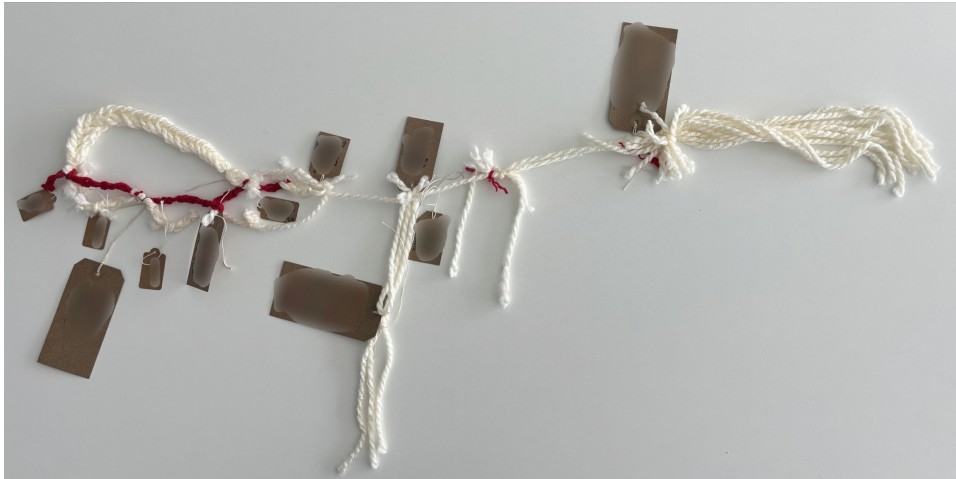

**Fig 5. LTA deconstruction and reconstruction (image a).**

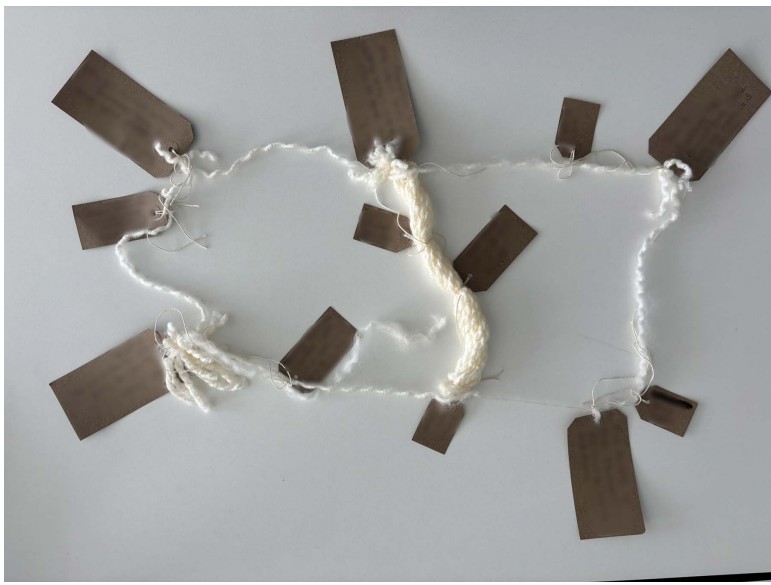

**Fig 6. LTA deconstruction and reconstruction (image b).**

While not everyone had a visceral reaction to the threads the meaning bestowed on the threads was highly symbolic for many family members and why some family members felt unable or uneasy about cutting or fraying their threads. For some cutting the threads was too finite, too definitive, and while the threads could be tied back together the act of breaking apart the threads symbolised a loss or change that was too painful to explore in this way (see examples of threads unable to be cut).

**Examples of threads unable to be cut:**

*"I couldn't cut it. I just kept tugging at it. That's what I tend to do, tug at it then I put it into a ball. And I think it's because I didn't like that fact that the family were split and he wouldn't have wanted that"* (Fig 8)

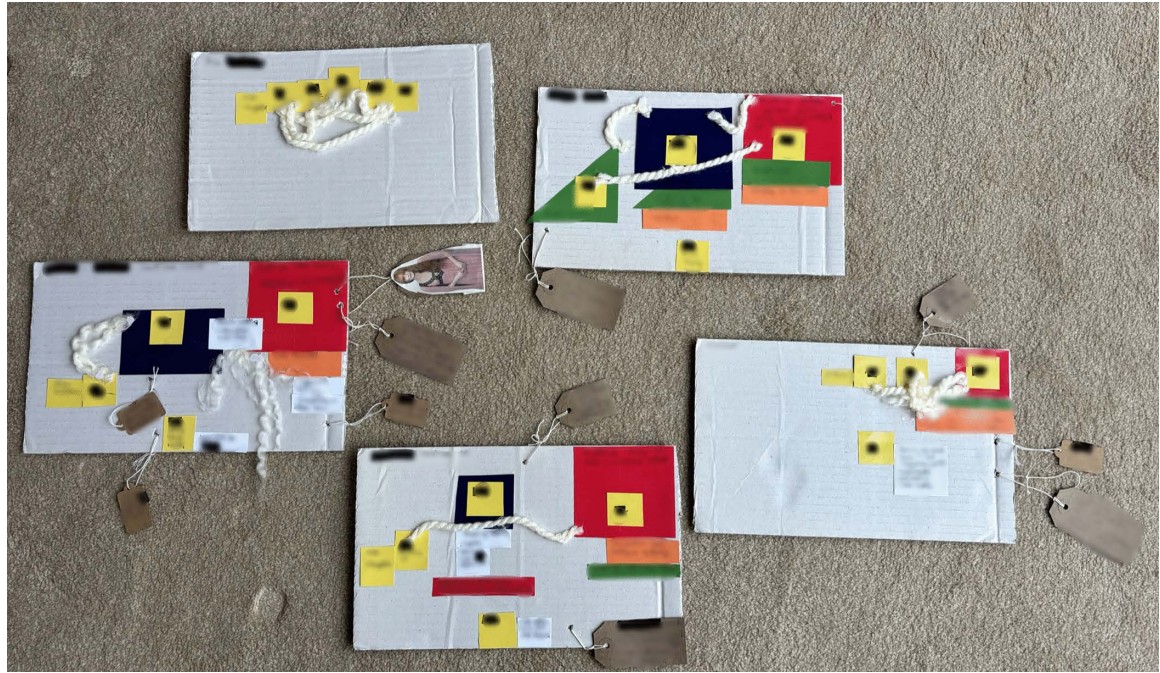

**Fig 7. LTA deconstruction and reconstruction (image c).**

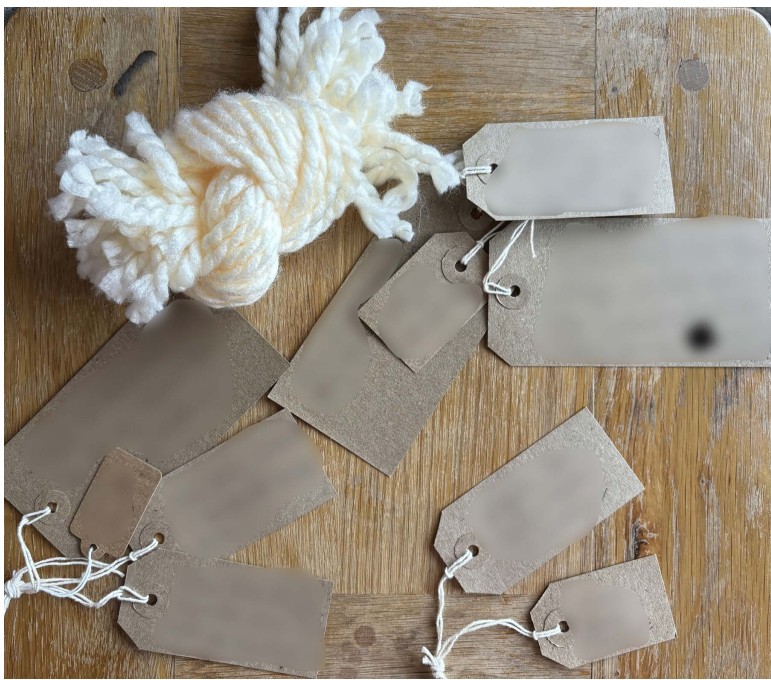

**Fig 8. LTA threads unable to be cut (image a).**

*"the first thing that jumped at me is, I can't pull any of this apart. This is how it should be, and I need to keep it like that which I have"* (Fig 9)

Most family members were able to interact in ways that mirrored the LTA guide in terms of cutting, fraying, tying. However, the meanings participants gave to their threads and the stories told about them were different and deeply personal (see examples of threads tied, frayed, broken). Not everyone felt able to follow the guide feeling constrained by the fact they only would reflect a moment in time rather than a more dynamic representation of their experience.

**Examples of threads tied, frayed, broken:**

*"I tied mine in a knot, and the knot to me represented the family and things like that. It was totally subconscious"* (Fig 10)

*"She is my best friend, so, that is why I've tied these three together to represent the strength that the three of us have. […] This one I just totally cut and this was about my ideas of what the perfect family was"* (Fig 11)

*"this bit here is my family so we've plaited that and it's six strings because it's me, my husband, the children and the brain injury because it lives with us. […] we've got all these extra threads here now ready to start and explore again"* (Fig 12)

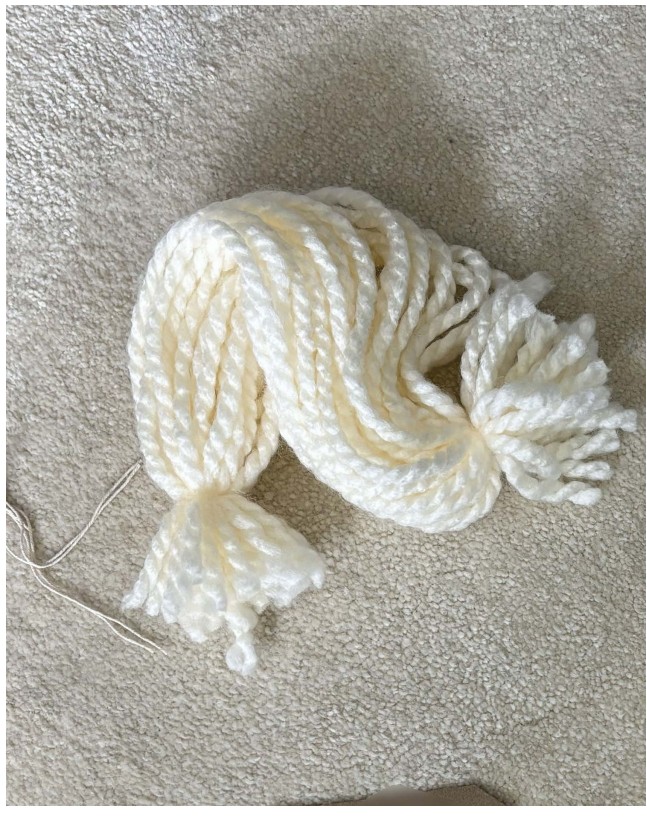

**Fig 9.  LTA threads unable to be cut (image b).**

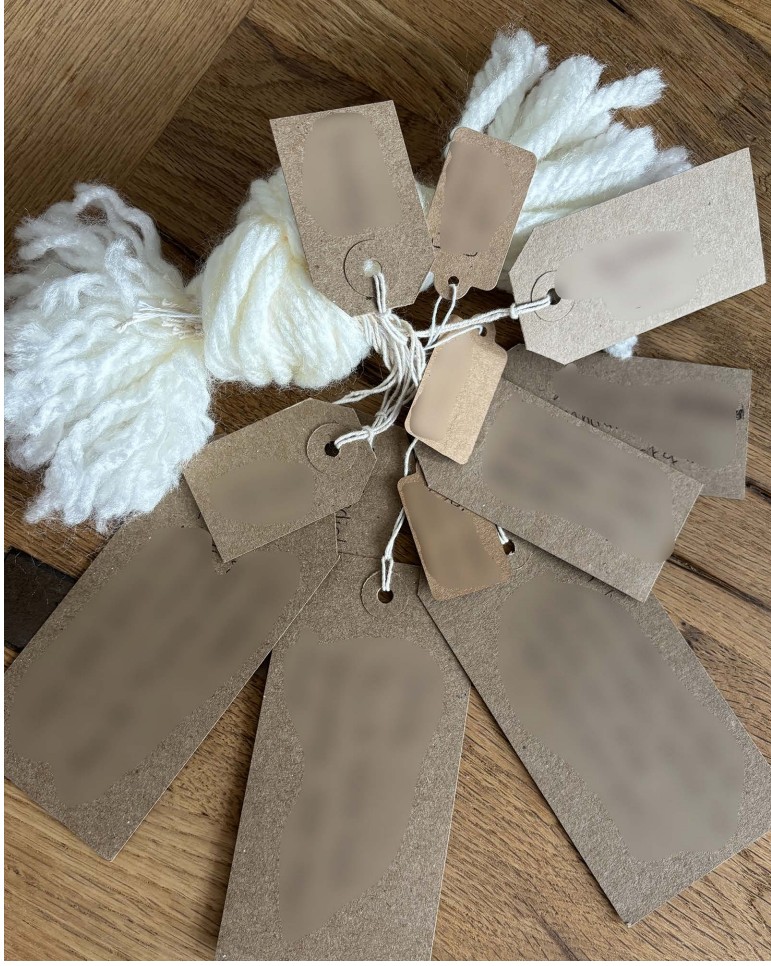

**Fig 10. LTA threads tied, frayed, broken (image a).**

*'This string here is one of my thinnest. I was on my own and it was during lockdown, and it was actually when he was going through a very intense form of post-traumatic amnesia'* (Fig 13)

*"the only thick mass that there sort of is, is the hope […] it was just hope for, I don't know, I suppose daily living tasks that we all take for granted and that was solid, solid the first three years"* (Fig 14)

*"The frayedness of just holding on really. […] I've tried to make it as frayed as possible because it is hard work now"* (Fig 15)

These acts of noticing, naming, and symbolism allowed family members to control their own narrative and position themselves within a story that had previously not been told.

**Steps toward understanding: "I've got to sort out a narrative" (Res4).** Family members acknowledged the difficulty of looking back over their experiences and expressed concern about wounds they might open. A sense of trepidation framed the initiation of this journey but there was some recognition that working through this could lead to benefit.

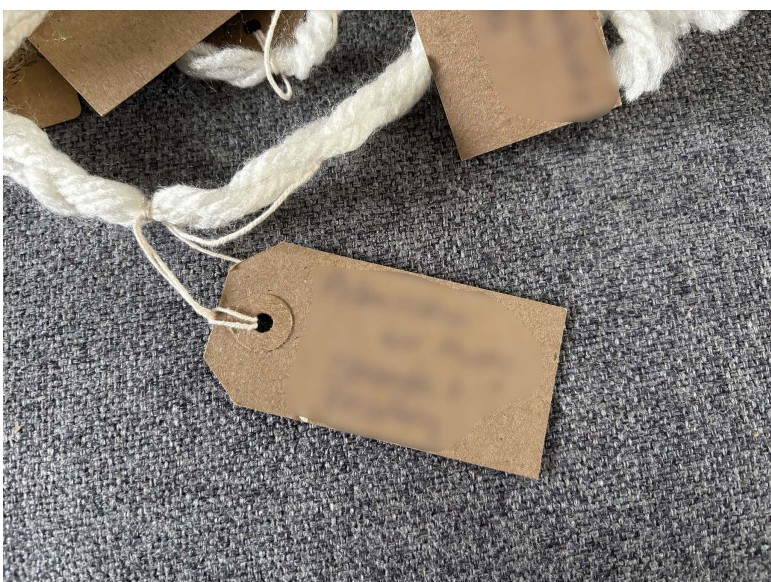

**Fig 11. LTA threads tied, frayed, broken (image b).**

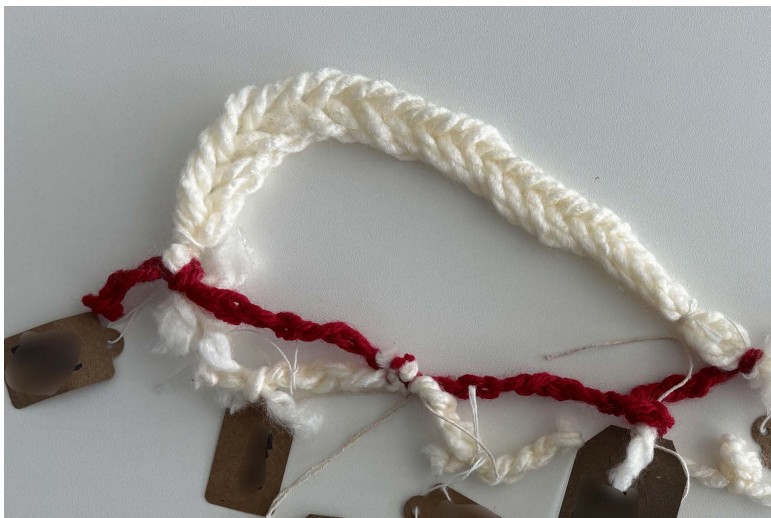

**Fig 12. LTA threads tied, frayed, broken (image c).**

*Res18 Int: "I didn't mean that we didn't look back. But look back with trying to give it a sense of perspective. You know, we do- you do look back. Even 19 years on, you look back to er… what you were expecting from life and how it's different […upset…]"*

*Res4 FG2: "it was the personalisation element of it. Because as soon as I had to make that step, that first step of wanting to do this because it's not a step I wanted to do, it was a step I really didn't want to do and I knew it would be painful and I was avoiding that pain"*

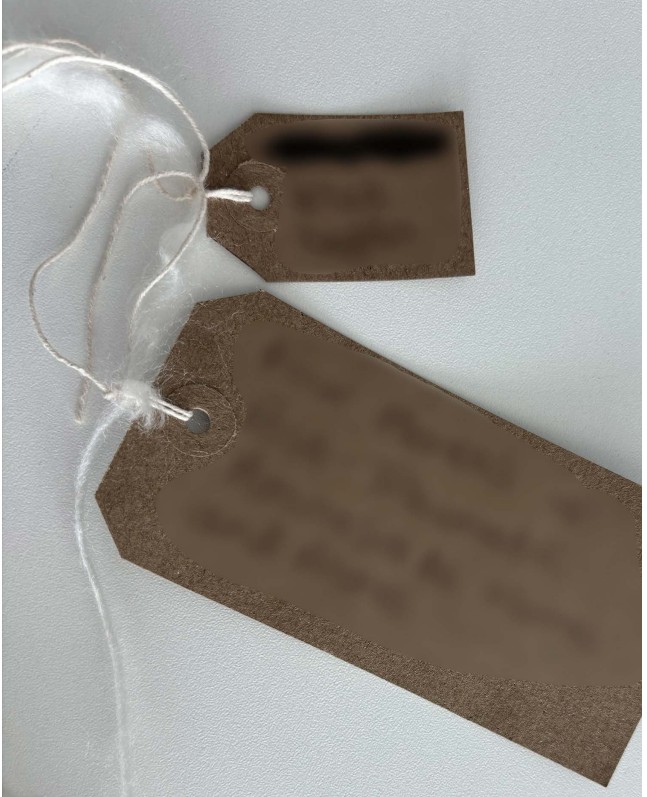

**Fig 13. LTA threads tied, frayed, broken (image d).**

Despite avoidance being a familiar stage of engagement, family members were often able to work through painful recollections and memories and then build perspective and balance into their story, challenge negative thoughts, providing an opportunity to rebalance, reframe, and re-evaluate. Family members created the opportunity to explore multiple interpretations of important facets of their story. One family member started with 'all the negatives' and then returned another day and used the back of the labels to counter these by finding the positives. However, for these family members, in the context of trauma, positive affirmations were often tinged with sadness, guilt and regret, captured in this comment from Res4 *"what would the purpose be? To learn humility, to experience pain, to… understand, like Jobe [laughs], 'I will test you', I don't know"*. Reflecting the complexity of life after TBI, family members enriched their story by expressing gratitude for deepened bonds, moments of joy, newfound strength, personal growth, affirmations for the future, and heightened insight into their own sense of self (see examples of using LTA to rebalance the narrative).

Res11 Int: *"I very much enjoyed working with the pack, it helped me realise how lucky we are that so many more threads could have remained broken and irreparable, so I've sort of, it made sense of it for me lots of threads not yet broken completely"*

**Examples of using LTA to rebalance the narrative:**

*"I ended up just putting like pluses and minuses as to how I felt it had impacted my life"* (Fig 16)

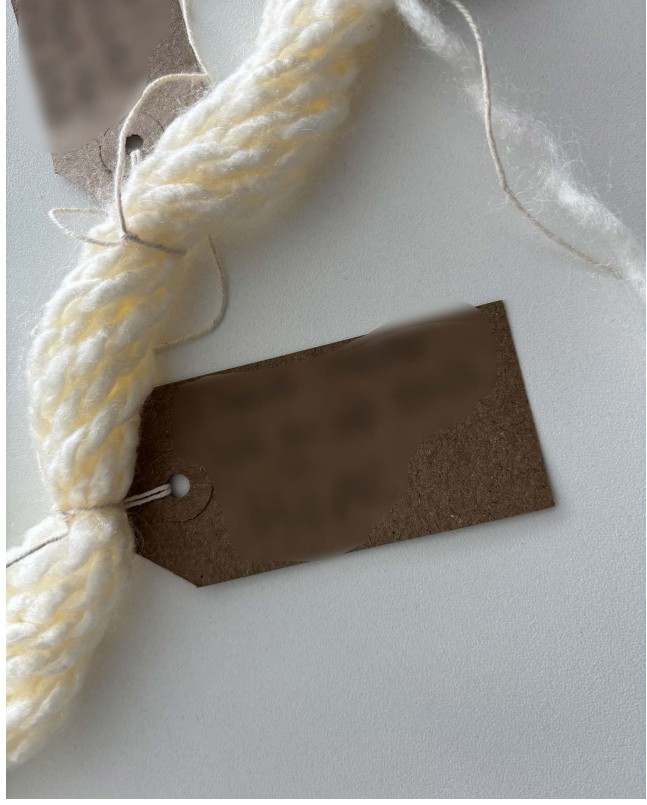

**Fig 14. LTA threads tied, frayed, broken (image e).**

*"So obviously, you know, there were a lot of negatives at the beginning […] about a week or so later, I looked and flipped it and then put some positives on, how things have got better."* (Fig 17)

*"Well, you bury all of the hurt, and you don't, you just don't want to get it out. You know, it's easier to just pretend that life is okay, you know"* (Fig 18)

Such enormous stories were sometimes hard to articulate, and this lack of narrative order also led to a lack of sense making.

*Res17 Int: "It's funny, so how I did this, right, I did it about 10 days ago or maybe more and wrote everything down in a jumble. […]I thought, God, I don't really know how to approach it with this particularly in that way, but I can understand connections, you know. […] Put it all back in the box and didn't think about it for another 10 days or something. […] And then a few days ago, I had started again"*

The rebalancing, enrichment and order was interpreted as steps to understanding because it allowed family members to move from a story saturated by trauma and loss to a story that facilitated an opportunity for hope and light.

### Theme 4: Creating the conditions for stories to be told

This final theme reflects the importance of **'safe spaces'** with family members **'bearing witness'** to the events they and others had experienced and the **'enhanced storytelling'** that was made possible through the LTA specifically.

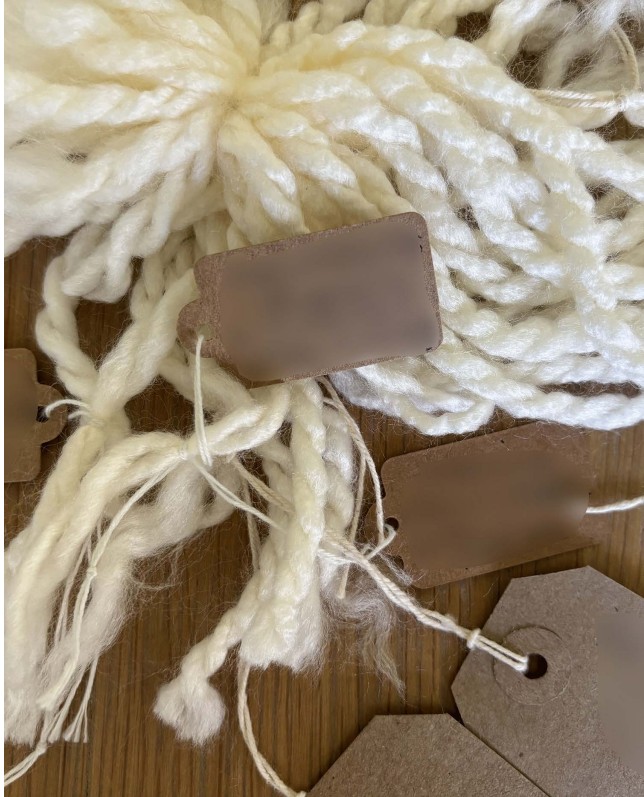

**Fig 15. LTA threads tied, frayed, broken (image f).**

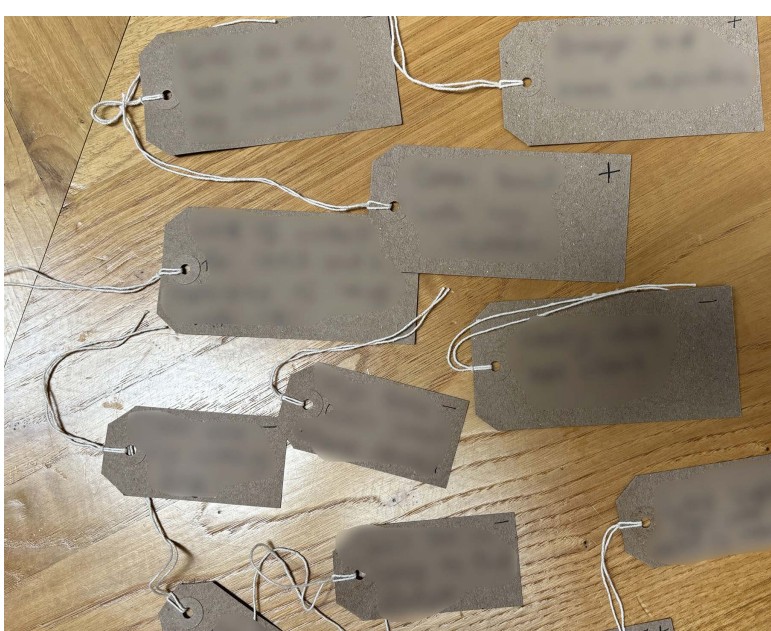

**Fig 16. LTA to rebalance the narrative (image a).**

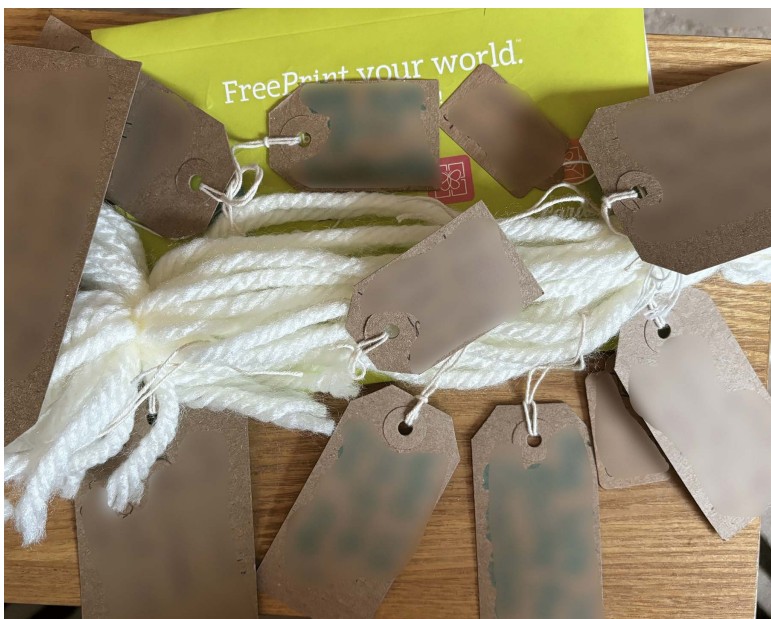

**Fig 17. LTA to rebalance the narrative (image b).**

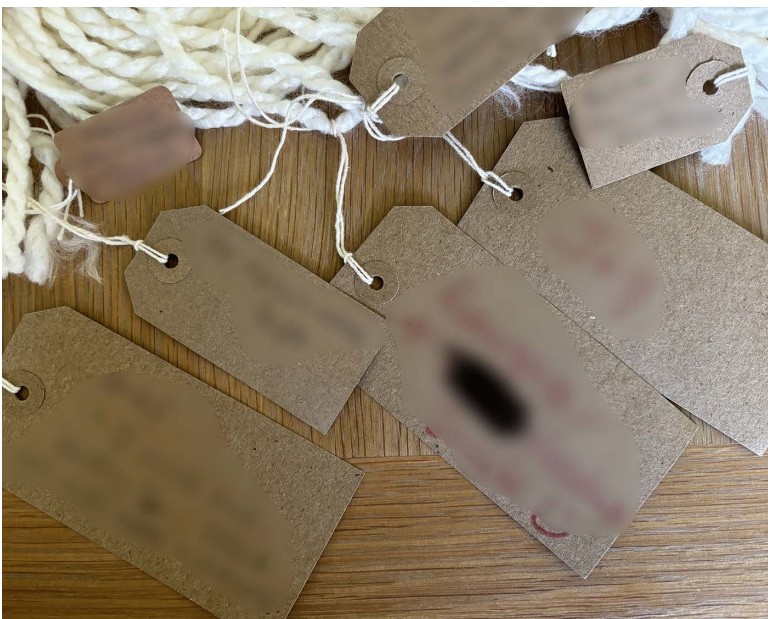

**Fig 18. LTA to rebalance the narrative (image c).**

**Safe spaces: "I'm not a creative person at all" (Res2).** Embedding the LTA within a safe space was critical to the ability to return to a difficult experience and engage in an open and curious way. In response to being introduced to the LTA some family members expressed scepticism and confusion with initial concerns centring on the lack of prescribed steps and the perceived need for creativity creating barriers to engaging.

*Res1 FG2: "I'll be honest, I was a bit confused as to what to do and where to go with it"*

*Res19 FG2: "I found the wool very nice to handle and very comforting. But I couldn't really work out how to do it"*

*Res6 FG2: "I was a bit worried that I might not have done it right. And I know you're like oh, there's no right or wrong answer, but there's a definite perfectionist hiding inside of me, like "are you absolutely sure that's the case!"*

In contrast others immediately resonated with the metaphor of threads, the value of storytelling as a therapeutic intervention and were excited to receive the LTA materials and make these their own. One family member said of the LTA that it seemed a 'pleasant way of starting' (Res11) emphasizing the welcome nature of a gentler way to probe the complexity of adjustment for families after TBI.

*Res10 FG1: "It looks really interesting. I think this is not something that can be measured on a scale or documented in any sort of algorithm or format that's scientific I suppose so there has to be a way in which you capture data that is interesting and still useful."*

Some family members felt that seeing examples of the threads may have helped them to overcome some of this initial hesitancy. While others expressed some concern that this may stifle or inhibit their own.

**Bearing witness: "no one's ever asked me how I was" (Res6).** Creating a space in which family members could explore their own sense of self post-TBI revealed the importance of telling their story and having this story heard in a way that acknowledged its truth and significance. Having spent so much time being unheard and unvalued, an opportunity to tell their story felt special and important.

*Res6 FG2: "you came all that way to see me and to listen to me and that felt, it felt warranted, that was really special actually"*

In this research the LTA was enhanced through the opportunity to both meet individually and within a group setting. With the interviews allowing participants the opportunity to tell their story in full, from their perspective rather than in sound bites of the injured person's survival and subsequent recovery.

*Res13 FG2: "It's nice to focus on your story and it's often quite difficult to do that and often we don't, or I don't realise where I sit within that story sometimes so it was helpful to look at it from that perspective. It was equally difficult to say it from- As my story, because I think that's why I've struggled quite so much with it as well because I was desperately trying not to tell it as my husband's story."*

*Res4 FG2: "I was a bit flummoxed by the question, What's the story from your point of view […] and I was a bit sort of, Oh well that's a totally new angle."*

Bearing witness was also about holding a space for others, listening to their story, connecting, offering support and insight, resonating with different yet similar events and circumstances. This lived experience meant that no feelings were judged, no thoughts dismissed, providing a space for these to be recognized and normalized. This need to find people who could resonate with their own was very strong for some.

*Res7 FG2: "I think also is that it can give a bit of a stress relief because your reaction sometimes to that partner or that person makes you quite angry and then you feel guilty for feeling angry because they're the injured one, whereas you can come away and you can say to other people who'll understand exactly how you feel."*

The opportunity to further explore their story and have this heard by others was a meaningful act of empathetic connection. In this shared space the family members felt validated, bringing feelings to the fore that had been felt but not shared.

**When words are not enough: "we've been through all of this trauma, and you're giving me some string?"** **(Res8).** At its simplest, the LTA was just wool in a box, which for some initially prompted doubt about whether something so simple could be helpful and risked trivialising the trauma they had been through. However, the LTA proved to be a 'way in' to their experiences that had not been available to them before. The chance to talk directly, for some for the first time, about their own story and not specifically that of the injured person was described by one family member as 'a revelation' (Res5). Several participants said that they did not think they would have shared such an intimate story without the LTA. It was the act of holding the threads that for many felt different, grounding them, providing comfort and connecting them in a visceral tangible way to their story.

*Res9 FG2: "I got it out last night, and, you know, we've had a bit of a rough time with <husband's> cancer and all the rest of it. So it is quite useful to hold on to those. It's actually quite comforting, actually, just to hold it."*

*Res11 Int: "also I think what was the nice thing as well was that this is so soft and I just, I don't know, I just felt like it was like a little comforter."*

*Res13 Int: "So I really liked this- As in, it was something that you could hold and- I found that quite helpful."*

Where the LTA appeared to draw its strength was it flexibility to support family members in a non-directed way

*Res6 FG2: "But I do think it's having almost a boundary, a picture frame, so that you can put some stuff inside it."*

This allowed family members to construct their story with people, places, experience and emotions that were deeply personal to themselves.

## Discussion

This study explored the potential of the LTA as a novel, narrative-based intervention to support family members following TBI. Through in-depth qualitative inquiry, we examined whether storytelling using the LTA could support subjective well-being and adjustment, how family members used the approach to understand the impact of TBI on themselves, and what uncertainties remain regarding its clinical application. The answers to the study objectives were intertwined throughout the reflexive themes which are discussed below. This includes addressing the final objective; to understand how the findings reflect the four dimensions of subjective experience described by Whiffin et al. [33].

### Understanding the context: Why families are vulnerable

The first theme, *Scaling cliffs with broken wings*, provided essential context for understanding the emotional and psychological landscape in which the LTA was introduced. The evidence in this theme reflects the negative outcomes of this population and an evidence base that is saturated by 'burden of care' [45]. As in previous studies, the participants described profound trauma, grief, loss, fatigue, marginalisation, isolation, discrimination and dehumanisation [15,46–53]. This mirrors the evaluation of change captured in the narrative dimensions of 'displacing and anchoring':

*Anchoring and displacing narratives were located across all timepoints, across relationships, and moved fluidly from displacing to anchoring and back again. They were used by the family to evaluate change and the impact of TBI on their lives* [33]*.*

Within these processes, the challenges of negotiating a sense of loss, with the ongoing presence of the person was highlighted, in keeping with the literature on 'ambiguous loss' after ABI [46,54,55]. These findings also converge with the clinically-grounded Family Tasks Model [31]. This way of conceptualising the family adaptation process highlights tasks including grieving mixed with trauma, restructuring expectations and roles, identity development and growth. These tasks or processes are not linear or mutually exclusive and shift dynamically over time depending on family life cycle and context, consistent with our findings which indicate complex and nuanced processes related to the negotiation of ambiguous loss alongside trauma and positive meaning making.

While variables such as injury severity, level of education, cognitive and emotional sequalae and loneliness, can negatively affect carers and families, it remains difficult to predict this with certainty [56]. Therefore, it was important in this study to recognise and present the specific lifeworld contexts of participants, i.e., their 'phenomenological field' [57], to make sense of their individual responses and demonstrate sensitivity, empathy and contextually bound experiences. This holds as true for taking family members through any type of supportive intervention, as it does within the context of these research findings.

This backdrop of cumulative distress and systemic neglect shaped how some participants approached the LTA, such as with caution and emotional fatigue, and may account for why carer outcomes are often worse over time not better [58–61]. These findings underscore the need for interventions that validate and centre the experiences of family members, not just as caregivers but as people in their own right who are navigating complex trauma and suffering [50].

## The need for support: Why interventions like the LTA matter

Despite the long-held understanding that carers are an at-risk population there is less understanding about the most effective way to support them [62]. In this study the second theme, *An entanglement of wellbeing*, highlighted the self-sacrificing roles family members assumed, often at the expense of their own health, wellbeing and identity. Despite a study showing carer burden is higher where carers report giving up one or more of their own activities to care for the survivor [63], carers are well-known to cut back on activities beneficial to their health and find it too difficult to take a break from their caring responsibilities [64]. This sense of intertwined wellbeing and invisible work is reflected in the 'Rupturing and stabilising' narrative dimension told about everyday family life:

*Rupturing narratives included difficulties in the family; [whereas] stabilising narratives included the work involved in a reducing conflict, bringing harmony and sustaining or redefining relationships* [33].

A study by Eady et al [65] found that patients and family members perceived family involvement in rehabilitation after TBI as critical, while some health professionals viewed it as unnecessary. In their review of social cognition in individuals with acquired brain lesions, Maggio et al. [66] imply that rehabilitation programs often overlook the inclusion and support of families in managing social cognition-related challenges, which may contribute to increased caregiver burden. Similarly, Culley et al [53] reported family members being ignored or excluded in medical and social contexts. In contrast, Morrison et al. [11] helps clarify the role of family in promoting positive rehabilitation and participation outcomes in the ABI community by demonstrating that family functioning remained a significant predictor of rehabilitation barriers, even after controlling for age, education, financial support, and community participation satisfaction.

Therefore, when service models lack a family-inclusive approach, they may inadvertently reinforce the perception that family members are not important, either to the survivor's outcomes or their own. This marginalisation is mirrored in the

research literature, where carer outcomes may be overlooked despite their central role in the survivor's rehabilitation and recovery [62]. Whiffin et al.'s [33] meta-synthesis showed how marginalisation can exacerbate isolation, and this increases the imperative to find connection.

*Isolating and connecting narratives [are] told about 'the space between'. This metaphorical and physical space between family members and those both internal and external to the family served to either bring people together or push them away* [33].

Being isolated is this way can mean family members struggle to get the help they need [33]. As a result, family members may come to de-prioritise their own health and wellbeing, contributing to a pattern of diminished self-worth and unmet need.

### The power of story: What the LTA offers

The third theme, *Hear me, see me: The power of story*, directly addressed the primary objective of this research and described, as well as showed, if and how the LTA was, or could be a useful strategy to support wellbeing. This theme also speaks to objective IV about how family members used the LTA to understand the impact of TBI on themselves and their family. This objective responds directly to the lack of studies that explore factors associated with positive outcomes after TBI [45].

In addition to the explicit statements about perceived benefits, participants found the LTA to be a powerful tool for reflection, meaning-making, and emotional processing. Participants used the LTA to understand the impact of TBI on themselves. For many this was the first time they'd had the opportunity to do this since their family member's injury. In this sense the LTA facilitated agency, voice and their own narrative identity.

*Narrative identity is a special kind of story—a story about how I came to be the person I am becoming…[…]…The life story also integrates life in a diachronic sense, that is, over time, ideally showing how the self of yesterday has become the self of today, the very same self that hopes or expects to become a certain kind of (different but still similar) self in the future* [67 p. 364].

The LTA enabled participants to shift their narrative from fragmented, trauma-saturated memories to more coherent, hopeful stories. This aligns with Whiffin et al.'s [33] final narrative dimension of harming and healing seen as *a temporary position for viewing life*:

*Harming and healing narratives: Harming narratives reflected the darkness that families lived with in their lives. The inability to process their experiences in a meaningful way often meant family members were left without hope, a sense of deep sadness, and an inability to start life again. They were founded on displacing, rupturing and isolating narratives. In contrast, healing narratives were told about the 'light', the move toward meaning, sense making, hope, personal growth from tragedy and moving forward* [33, p.15].

LTA provided a flexible but explicit process by which family members could engage in this natural and human process of meaning making and narrative construction, i.e., an active means of co-constructing identity through this interaction [68]. Such outcomes are indicative of a values-based approach that has potential to humanise research and practice [69].

The findings highlight the spontaneous way in which participants used the materials to help move towards, identify, and label or name their experiences as they developed their stories. Although the LTA was devised without reference to a specific therapeutic modality, the process of noticing and labelling in this way is a common feature of contemporary evidence-based therapeutic approaches including CBT [70], Mindfulness Based Cognitive Therapy [71] and Acceptance

and Commitment Therapy [72]. The flexibility of the LTA process also provided an opportunity for families to face and articulate trauma or loss previously avoided, renegotiation of roles and identity and for some to identify new, future-facing narratives, in their own way and at their own pace. Again, there is close alignment with the Family Tasks Model. This conceptualisation of the existential 'work' required by families [73] and of the dynamic, context-bound and complex nature of at times conflicting emotions and processes contrasts with more commonly implemented linear, structured intervention approaches based on education and emotional support [74]. Whilst such approaches might be relevant and helpful, where families have yet to reflect on their own circumstances or where there is fear of doing so, the LTA has potential to offer a novel 'way in' to these complex existential tasks related to the adaptation process. Inherent in giving the LTA materials to families to make use of them in their own way is a trust in family capacity to be resilient or find their own solutions to challenges, as reflected in the principles of systemic family therapy [31].

## Creating the conditions for storytelling: Clinical considerations

The final theme, *Creating the conditions for stories to be told*, addressed some of the uncertainties around clinical application, examining how family members engaged with the LTA, and as such met objectives II and III. While the LTA was broadly acceptable and meaningful, its success depended on several factors: the creation of safe, non-judgmental spaces; the flexibility to engage in one's own way and at one's own pace; and the opportunity to share stories with others who understood. Although self-exploration was important this was often impeded by guilt, exhaustion, and a deeply ingrained prioritisation of others. These findings mirror research with carers in other situations such as life limiting illness [75] and chronic fatigue [76] who also found self-compassion, self-care and acceptance difficult. Therefore, the LTA was not an activity participants felt able to undertake alone and the need for a process and support from others was paramount. These findings echo those of a recent study by Drake et al [77] who showed the content of a video-based resource for parents of a child with brain injury could be emotionally *'triggering'*. The study highlighted that whilst some parents might require additional support, for others this approach provided them with control over the pace at which they approached the enormity of making sense of their child's and their own changes. Our findings resonate, further indicating the usefulness of family members being able to engage with the materials at their own pace, allowing a 'soft entry' or 'gentle step' [77 p. 11] into the potentially distressing process of making sense of their own experience. The potential utility of creative, arts-based approaches that we anticipated when developing the project is also highlighted. Previous qualitative work in the field of ABI and caregiving has employed creative methods such as photovoice [78] or 'wool and stones' [32] to enable participants to access and share metaphorical or embodied meanings in their daily experiences. Similarly, here we have extended these ideas and principles beyond the research context as reflected to an extent in the arts and health movement [34,79].

While there are no effective standardised interventions employed to support family members after brain injury [74], peer support is often regarded as beneficial in such groups and features commonly as an aspect of family intervention [74]. However, in this study both the group and specific individual components were valued, and future implementation should consider how to balance elements of self and supported styles of facilitation. Findings also suggest that any intervention aimed at family members must be sensitive to these dynamics and offer permission, space, and support for self-exploration, whilst recognising that family members might be fatigued and reluctant, finding self-care and self-exploration a difficult process.

Findings suggest that the LTA is not a standalone intervention but one that requires thoughtful facilitation. Perceived benefit of the LTA likely varies depending on multiple contextual aspects relating to the background and experiences of the family member including history of services and support, culture, religion and ethnicity, mental health or service-related needs and individual characteristics such as comfort with using creative materials. As such the LTA aligns closely with the notion of a complex healthcare intervention as defined by the UK Medical Research Council's guidelines [80]. In keeping with this guidance, and based on the findings of our study, a systems approach to evaluation rather than a theory-based,

efficacy and effectiveness-based approach should be taken in future work. The approach taken to this initial pre-feasibility phase has highlighted several aspects of family and service context that likely influence delivery of, and response to, the LTA. However, further detailed delineation of these context-bound processes and optimal facilitation are required. This indicates that future work will necessarily involve study of the approach in more diverse contexts in close collaboration with family members, and wider stakeholders which is commensurate with a values framework for qualitative research [69]. Table 9 summarises our understanding in response to objective II and III while offering areas for future consideration.

### Strengths and limitations

As an in-depth exploratory study, this research has allowed us to both examine the potential for the LTA to be a therapeutic intervention, while also honouring the priorities of participants by highlighting the unique vulnerabilities and complex needs of families after TBI, which were central to their experiences and perspectives. As such, the answers to the research objectives were not as clear-cut as those typically yielded by pre-feasibility studies and reflects a more contextually bounded qualitative insight. However, this leaves opportunity for further analysis and closer scrutiny of our understanding of the complex lives of families after TBI and specifically, validation of the narrative dimensions as a framework in which to view these.

Furthermore, despite the unique characteristics of study participants, the final demographics of this sample mean that the findings reflect experiences constructed through a predominantly White English, female, heteronormative lens. While

**Table 9. Summary of understanding reached regarding feasibility and acceptability of the LTA as a therapeutic tool.**

| | Understanding reached | Areas for future exploration |
|---|---|---|
| Acceptability | While most participants engaged meaningfully, some found the metaphor of threads confusing or emotionally overwhelming. | Clearer guidance and optional examples may support engagement without constraining creativity. |
| Adherence | The lack of explicit instructions was an important part of the LTA and most participants were able to engage as suggested in the LTA guide. However, emotional readiness varied widely. | The LTA materials would benefit from further development to further facilitate individual engagement. |
| Level of facilitation | Participants were able to engage in the LTA without facilitation. However, the follow-up interview was key in enhancing their understanding and their story construction. The skills of the facilitator were critical during these interviews. The lead researcher's expertise in unstructured interviewing likely enhanced the depth of engagement. We therefore do not recommend use of the Life Threads approach as a standalone tool. | Future studies should explore what training and support facilitators need to replicate these outcomes. |
| Representative recruitment | NHS settings have limited contact with patients and families by two years. Marginalised groups were underrepresented in the study. | How to reach and recruit under-researched groups. How to recruit family members discharged from the NHS and not known to community services. |
| Choice of primary outcome measure | The key outcomes appeared to be a recognition of own narrative, shift in identity narrative and improved personal agency leading to greater sense of wellbeing. | A scoping review of measures used to assess outcomes after TBI for families may be useful. However, these findings must be explored with PPIE groups to understand the relevance of these sense of self, narrative change and wellbeing. |
| Mode of delivery | The LTA was enhanced by both one to one and group support with a facilitator who shows flexibility, empathy and responsiveness. | To understand the different modes of delivery and how these reflect the needs of different subgroups, e.g., parents, spouses, siblings |
| Comparator arm | A traditional trial design may not be appropriate for the LTA due to the heterogeneity of both the population and their needs. | Researchers may want to consider participatory co-design methods to facilitate a more flexible mode of inquiry. |

some injured persons required complex care interventions these perspectives reflect the experiences of family members whose injured relative more often experienced severe injuries with outcomes that allowed them to live relatively independent lives. It is possible that participants in this study had higher capacity for engagement in rehabilitation or had more open communication styles. While our PPIE group reflected greater gender and ethnic diversity and viewed the LTA as culturally acceptable, how the LTA could be beneficial for a more diverse sample or those with more severe outcomes post-injury remains unanswered.

The replication of previous findings regarding the challenges, trauma and re-traumatisation experienced by families, many of whom feel neglected or rejected by services and society, highlights the pressing need for family-centred approaches to neurorehabilitation [81,82] and families who should be offered supportive interventions in their own right [17,83]. Whilst we have identified the clinical potential of the LTA, the nature and scale of this study limit the extent to which we can recommend its wider implementation. Therefore, we have established the Life Threads–TBI Network through Anchor Point (www.anchorpointabi.org/research) as a commitment to ongoing dialogue with professionals, family members and researchers regarding values-based, family-centred approaches, and to further refine the LTA for future research.

## Conclusion

The primary aim of this study was to understand the clinical potential of storytelling through the Life Threads Approach and gather the information required to plan a feasibility randomised control trial. The findings of this study demonstrate that the LTA offers a meaningful way for family members to explore, express, and make sense of their experiences following TBI. The LTA supports narrative reconstruction, fosters connection, agency and provides a rare opportunity for self-reflection. However, its implementation must be trauma-informed, paced, and supported by skilled facilitators who can hold space for complex emotional responses. While not a panacea itself, the LTA represents a promising step toward more inclusive, humanising support for families navigating the long-term consequences of TBI and offering a more 'gentle step' towards meaning making and adaptation than offered by more formal, structured, information-based approaches. Despite a traditional clinical trial not being the most appropriate research design to further understand the potential benefits of such an approach, a context sensitive participatory co-design may yield more important insights.

## Supporting information

**S1 Table. Response to Braun and Clarke's (2022) 15-point checklist for 'good' reflexive TA.** This is a completed checklist to ensure reflexive TA was conducted rigorously.
(DOCX)

**S2 Table. Reflexive Thematic Analysis Reporting Guidelines.** This is a completed checklist commensurate with reflexive TA to ensure all relevant methods are reported.
(DOCX)

**S3 Table. Response to Tracy's big tent criteria.** This is a reflexive account of rigour and credibility in this study.
(DOCX)

## Acknowledgments

We would like to sincerely thank our PPIE advisory group members: Wal Warmington, Julie Berry, Leroy Jones, John Kennedy and the staff at QMC, Nottingham University Hospitals NHS Trust; Royal Derby Hospital, University Hospitals of Derby and Burton NHS Foundation Trust and Queen Elizabeth Hospital, University Hospitals Birmingham NHS foundation Trust. We are also indebted to the engagement of regional Headway groups and branches in addition to Headway UK,

UKABIF, BABICM, Anchor Point, Cygnet and Derbyshire Carers Association. We are particularly grateful to Dr Brandon Smith for ongoing support in the preparation of this manuscript.

## Author contributions

**Conceptualization:** Charlotte Jane Whiffin, Caroline Ellis-Hill, Alyson Norman, Jo Clark-Wilson, Fergus Gracey.

**Formal analysis:** Charlotte Jane Whiffin, Caroline Ellis-Hill, Fergus Gracey.

**Funding acquisition:** Charlotte Jane Whiffin, Caroline Ellis-Hill, Alyson Norman, Morag Lee, Parmjeet Kaur Singh, Mark Holloway, Jo Clark-Wilson, Natasha Yasmin, Sara Rose, Fergus Gracey.

**Investigation:** Charlotte Jane Whiffin, Caroline Ellis-Hill, Alyson Norman, Mark Holloway, Natasha Yasmin, Sara Rose, Fergus Gracey.

**Methodology:** Charlotte Jane Whiffin, Caroline Ellis-Hill, Alyson Norman, Mark Holloway, Natasha Yasmin, Fergus Gracey.

**Project administration:** Charlotte Jane Whiffin, Morag Lee, Parmjeet Kaur Singh, David Sheffield, Fergus Gracey.

**Writing – original draft:** Charlotte Jane Whiffin, Caroline Ellis-Hill, Alyson Norman, Fergus Gracey.

**Writing – review & editing:** Charlotte Jane Whiffin, Caroline Ellis-Hill, Alyson Norman, Morag Lee, Parmjeet Kaur Singh, Mark Holloway, Jo Clark-Wilson, Natasha Yasmin, Sara Rose, David Sheffield, Fergus Gracey.

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
