## [Decision Letter · Decision Letter 0]

13 Jan 2026

PONE-D-25-43207Supported storytelling through the ‘Life Threads’ approach for family members after Traumatic Brain Injury: ‘We've been through all of this trauma, and you're giving me some string?’PLOS One

Dear Dr. Whiffin,

Thank you for submitting your manuscript to PLOS ONE. After careful consideration, we feel that it has merit but does not fully meet PLOS ONE’s publication criteria as it currently stands. Therefore, we invite you to submit a revised version of the manuscript that addresses the points raised during the review process.

We look forward to receiving your revised manuscript.

Kind regards,

Kamalakar Surineni, MD, MPH

Guest Editor

PLOS One

Journal Requirements:

“This project is funded by the National Institute for Health and Care Research (NIHR) under its Research for Patient Benefit (RfPB) Programme (Grant Reference Number NIHR204092). The views expressed are those of the author(s) and not necessarily those of the NIHR or the Department of Health and Social Care.”

4. In the online submission form you indicate that your data is not available for proprietary reasons and have provided a contact point for accessing this data. Please note that your current contact point is a co-author on this manuscript. According to our Data Policy, the contact point must not be an author on the manuscript and must be an institutional contact, ideally not an individual. Please revise your data statement to a non-author institutional point of contact, such as a data access or ethics committee, and send this to us via return email. Please also include contact information for the third party organization, and please include the full citation of where the data can be found.

5. We note that this data set consists of interview transcripts. Can you please confirm that all participants gave consent for interview transcript to be published?

If they DID provide consent for these transcripts to be published, please also confirm that the transcripts do not contain any potentially identifying information (or let us know if the participants consented to having their personal details published and made publicly available). We consider the following details to be identifying information:

- Names, nicknames, and initials

- Age more specific than round numbers

- GPS coordinates, physical addresses, IP addresses, email addresses

- Information in small sample sizes (e.g. 40 students from X class in X year at X university)

- Specific dates (e.g. visit dates, interview dates)

- ID numbers

Or, if the participants DID NOT provide consent for these transcripts to be published:

- Provide a de-identified version of the data or excerpts of interview responses

- Provide information regarding how these transcripts can be accessed by researchers who meet the criteria for access to confidential data, including:

a) the grounds for restriction

b) the name of the ethics committee, Institutional Review Board, or third-party organization that is imposing sharing restrictions on the data

c) a non-author, institutional point of contact that is able to field data access queries, in the interest of maintaining long-term data accessibility.

d) Any relevant data set names, URLs, DOIs, etc. that an independent researcher would need in order to request your minimal data set.

For further information on sharing data that contains sensitive participant information, please see: https://journals.plos.org/plosone/s/data-availability#loc-human-research-participant-data-and-other-sensitive-data

If there are ethical, legal, or third-party restrictions upon your dataset, you must provide all of the following details (https://journals.plos.org/plosone/s/data-availability#loc-acceptable-data-access-restrictions):

a) A complete description of the dataset

b) The nature of the restrictions upon the data (ethical, legal, or owned by a third party) and the reasoning behind them

c) The full name of the body imposing the restrictions upon your dataset (ethics committee, institution, data access committee, etc)

d) If the data are owned by a third party, confirmation of whether the authors received any special privileges in accessing the data that other researchers would not have

e) Direct, non-author contact information (preferably email) for the body imposing the restrictions upon the data, to which data access requests can be sent

Reviewers' comments:

Reviewer's Responses to Questions

**Comments to the Author**

1. Is the manuscript technically sound, and do the data support the conclusions?

Reviewer #1: Yes

Reviewer #2: Yes

2. Has the statistical analysis been performed appropriately and rigorously? 

Reviewer #1: N/A

Reviewer #2: No

3. Have the authors made all data underlying the findings in their manuscript fully available?

Reviewer #1: Yes

Reviewer #2: No

4. Is the manuscript presented in an intelligible fashion and written in standard English?

Reviewer #1: Yes

Reviewer #2: Yes

5. Review Comments to the Author

Reviewer #1: 1. Theoretical Framework

The manuscript would benefit from a clearer articulation of the theoretical underpinnings of the Life Threads approach. While storytelling and narrative therapy concepts are implied, referencing key frameworks (e.g., narrative identity reconstruction, post-traumatic growth, meaning-making) would strengthen conceptual grounding.

Clarify how the creative arts component (the “thread” metaphor) integrates with established psychological constructs to promote recovery and resilience.

2. Methodological Transparency

The methods are consistent with qualitative rigor, but additional detail is warranted:

Specify participant recruitment strategies and inclusion/exclusion criteria (e.g., relationship to TBI survivor, time since injury, level of functional impairment).

Elaborate on data collection — for example, duration and format of sessions, interviewer background, and reflexivity considerations.

Describe the analytic process more thoroughly: How were themes generated, reviewed, and validated among authors? Was data saturation achieved?

Consider including a brief reflexive statement addressing the researchers’ positions and potential influence on interpretation.

3. Data Presentation and Credibility

The use of participant quotations is effective, but ensure each theme is consistently illustrated with verbatim excerpts from multiple participants.

In Table or Appendix form, provide a concise summary of major themes/subthemes and representative quotes to improve accessibility and transparency.

Clarify whether the quotes were edited for brevity or grammatical correction and how confidentiality was preserved.

4. Interpretation and Discussion

The discussion insightfully links findings to emotional adaptation and meaning-making, but could be deepened by connecting to broader literature on family identity reconstruction after TBI, ambiguous loss, and family systems recovery.

Address possible limitations in generalizability, as participants may represent families with higher engagement in rehabilitation or more open communication styles.

The phrase in the title (“you’re giving me some string?”) powerfully captures ambivalence; consider analyzing this further as an example of the initial skepticism families may feel toward non-traditional interventions and how that evolves into acceptance.

5. Ethical and Practical Implications

Expand briefly on ethical safeguards for emotionally sensitive interviews (e.g., availability of psychological support if participants became distressed).

Include implications for clinical implementation — for instance, how Life Threads could be integrated into multidisciplinary neurorehabilitation settings or adapted for other caregiver populations.

Some Minor Comments:

The title is evocative and engaging; ensure quotation marks are consistent with journal style.

The abstract should explicitly mention sample size, analytic approach (e.g., reflexive thematic analysis), and main themes.

Minor grammatical and stylistic adjustments are recommended (e.g., “effecting” → “affecting”; “participants’ voices were captured authentically”).

References to similar narrative or arts-based rehabilitation programs (e.g., creative reminiscence therapy, photovoice) would enhance context.

Verify formatting of quotations (indentation, punctuation) per PLOS ONE style guide.

Reviewer #2: The study is well-written and addresses a significant gap in long-term family support . The integration of Patient and Public Involvement and Engagement (PPIE) is a major technical strength; the authors demonstrate high-quality collaborative research by involving family members as co-investigators and advisory group members throughout the study lifecycle .

- Re-title the manuscript as a "Qualitative Proof-of-Concept" rather than a "pre-feasibility" study, as your data concludes that a traditional randomized control trial (RCT) is likely inappropriate for this intervention.

- The sample is 95% White and 85% female; please expand the discussion on how the metaphor-heavy "Life Threads" approach might be received by more diverse cultural groups or male family members.

- Explicitly address whether the approach can be a standalone tool, as the data indicates success relied heavily on skilled facilitation and expert-led unstructured interviews.

- Briefly detail the vetting process used to manage the 22% of expressions of interest identified as phishing attempts to ensure data integrity.

- Include any available insights into why three participants withdrew after the first focus group to better evaluate the intervention's "acceptability" .

- While the study follows Reflexive Thematic Analysis (rTA) standards, ensure the re-framing acknowledges the shift from quantitative feasibility to qualitative insight.

6. PLOS authors have the option to publish the peer review history of their article (what does this mean?). If published, this will include your full peer review and any attached files.

Reviewer #1: **Yes:** VENKATA VIJAYA K DALAI

Reviewer #2: No

---

## [Author Response · Author response to Decision Letter 1]

4 Mar 2026

We would like to thank both reviewers for their thoughtful and constructive feedback. We have carefully considered all comments and have revised the manuscript accordingly. Below, we provide a detailed, point-by-point response. Revisions requested by the editor are detailed in the cover letter. A 'response to reviewers' file has also been uploaded to the Editorial Manager with editorial and reviewer revisions summarised in a tabular format.

Reviewer 1

The manuscript would benefit from a clearer articulation of the theoretical underpinnings of the Life Threads approach.

Referencing key frameworks (e.g., narrative identity reconstruction, post-traumatic growth, meaning-making) would strengthen conceptual grounding.

Clarify how the creative arts component (the “thread” metaphor) integrates with established psychological constructs to promote recovery and resilience.

Thank you for this helpful suggestion. We have now strengthen the manuscript by adding further explanation of theoretical underpinnings of the LTA including narrative, identity reconstruction and post-traumatic growth. Further, we have clarified the thread metaphor and how this integrates with creative arts approaches. p.3-4

Specify participant recruitment strategies and inclusion/exclusion criteria (e.g., relationship to TBI survivor, time since injury, level of functional impairment).

We have updated Table 3 to include the exclusion criterion for this study.

We have added further detail on the recruitment process to ensure this is transparent. I can confirm there were no restrictions based on injury type, family role or level of impairment. We have clarified this in the manuscript. p.7 – 8

Elaborate on data collection — for example, duration and format of sessions, interviewer background, and reflexivity considerations.

Under the guidance of Braun and Clarke, we use the header ‘dataset generation’ to refer to data collection. However, we acknowledge that it may not be immediately clear that dataset generation refers to data collection. We have therefore added ‘data collection’ to the heading for clarity.

We can confirm that all details pertaining to the duration and format of the interviews and focus groups are described here.

Interviewer backgrounds are contained within a ‘positionality statement’ under the main header of analysis. p.9

Describe the analytic process more thoroughly: How were themes generated, reviewed, and validated among authors?

Thank you for requesting further information on this important step. We have now moved the information from the supporting file into the main manuscript so that our methods are transparent and explicit for readers. p.11-12

Was data saturation achieved?

You are right to raise the question of data saturation. However, in the analytical method we have used, saturation is contentious and Braun and Clarke have written extensively about the lack of alignment of data saturation with their reflexive approach to thematic analysis.

We have now added clarity on this point to the manuscript and signposted to a more appropriate concept of ‘data sufficiency’. p.8

Consider including a brief reflexive statement addressing the researchers’ positions and potential influence on interpretation.

We do have a positionality statement that addresses all of the researchers’ backgrounds. I have now added to this that our experiences within this field will have influenced how we interpreted the data. p.10

Ensure each theme is consistently illustrated with verbatim excerpts from multiple participants.

I have mapped the respondent quotes against the themes and identified those where no data were presented.

I have now added data from Res 15, 16, 18 and 20.

The manuscript now contains data from every participant at least once.

In Table or Appendix form, provide a concise summary of major themes/subthemes and representative quotes to improve accessibility and transparency.

I can confirm that Table 8 provides a useful summary of the themes and sub-themes.

Direct quotes are contained in this table; however these were not shown as such. We have now added quotation marks so that this is clear. Table 8

Clarify whether the quotes were edited for brevity or grammatical correction and how confidentiality was preserved.

Quotes used in the manuscript are de-identified verbatim extracts of raw data. We have now explained this in the manuscript. p.16

The discussion insightfully links findings to emotional adaptation and meaning-making, but could be deepened by connecting to broader literature on family identity reconstruction after TBI, ambiguous loss, and family systems recovery.

Many thanks for this important suggestion. We have now extended the debate around emotional adaptation and key constructs in the ABI literature including ambiguous loss and the family tasks model. We have also discussed the wider literature on family interventions after ABI. p.35 – 41

Address possible limitations in generalizability, as participants may represent families with higher engagement in rehabilitation or more open communication styles.

Thank you for identifying this important limitation. We have now added this to the strengths and limitation section of the manuscript. p.41

The phrase in the title (“you’re giving me some string?”) powerfully captures ambivalence; consider analyzing this further as an example of the initial skepticism families may feel toward non-traditional interventions and how that evolves into acceptance.

Thank you for this helpful suggestion. We have now alluded to this initial doubt in under the sub-title. p.34

Expand briefly on ethical safeguards for emotionally sensitive interviews (e.g., availability of psychological support if participants became distressed).

Our ethical safeguards for participant’s emotional wellbeing are now described as requested. p.5

Include implications for clinical implementation — for instance, how Life Threads could be integrated into multidisciplinary neurorehabilitation settings or adapted for other caregiver populations.

We have made it clear that we are not ready to recommend use of the LTA in clinical practice without further research. However, we do advocate for family centred rehabilitation and that families have access to supportive interventions in their own right. We have also signposted to the Life Thread Network for people interested in the approach (www.anchorpointabi.org/research) p.44

The title is evocative and engaging; ensure quotation marks are consistent with journal style.

I have reviewed the style guides for manuscript titles. I can confirm that the title meets the character limit. There does not appear to be clear guidance on the use of quotation marks. However, we have revised the title to use quotation marks specifically. p.1

The abstract should explicitly mention sample size, analytic approach (e.g., reflexive thematic analysis), and main themes.

We have made revisions to the abstract to ensure these methods are explicit. p.2

Minor grammatical and stylistic adjustments are recommended (e.g., “effecting” → “affecting”).

We have reviewed the manuscript and made adjustments to improve grammar and correct errors.

References to similar narrative or arts-based rehabilitation programs (e.g., creative reminiscence therapy, photovoice) would enhance context.

Thank you for suggesting this. We have reflected on the basis of the approach in creative arts in the introduction and have returned to related work in the discussion. Here we also recognise the potential for methods originating as research methods (such as photo elicitation and the ‘wool and stones’ approach) in offering creative opportunities for resources or interventions to help families negotiate the challenges of life after brain injury and to find new and adaptive meanings. Introduction/ Discussion

Verify formatting of quotations (indentation, punctuation) per PLOS ONE style guide.

I have reviewed the style guides for PLOS ONE and cannot see explicit guidance for the presentation of direct quotes. I have reviewed recently published qualitative manuscripts and there does not appear to be a consistent approach. The current manuscript is in line with current styles.

Reviewer 2 Re-title the manuscript as a "Qualitative Proof-of-Concept" rather than a "pre-feasibility" study, as your data concludes that a traditional randomized control trial (RCT) is likely inappropriate for this intervention.

Thank you for this suggestion which we have discussed at length. However, given that the published protocol and original funding were aligned to a qualitative pre-feasibility study we feel it is important to maintain this consistency. We do then highlight that the answers to the objectives were not as clear cut as those typically yielded in pre-feasibility studies in the strengths and limitations.

The sample is 95% White and 85% female; please expand the discussion on how the metaphor-heavy "Life Threads" approach might be received by more diverse cultural groups or male family members.

This is an important point and one that we explored in our PPIE group. We have added this insight into the manuscript. p.44

Explicitly address whether the approach can be a standalone tool, as the data indicates success relied heavily on skilled facilitation and expert-led unstructured interviews.

We have now added to Table 9, that we do not support use of the Life Threads approach as a standalone tool. Table 9

Briefly detail the vetting process used to manage the 22% of expressions of interest identified as phishing attempts to ensure data integrity.

Thank you for requesting clarification of this point. I have added further detail to the manuscript about this. p.12

Include any available insights into why three participants withdrew after the first focus group to better evaluate the intervention's "acceptability".

I have now added further insight into why these three family members felt unable to complete the LTA. However, this data was not collected formally. p.12-13

While the study follows Reflexive Thematic Analysis (rTA) standards, ensure the re-framing acknowledges the shift from quantitative feasibility to qualitative insight.

Thank you for this helpful recommendation. This has been added to the manuscript. p.43

---

## [Decision Letter · Decision Letter 1]

29 Apr 2026

Supported storytelling through the ‘Life Threads’ approach for family members after Traumatic Brain Injury: ‘We've been through all of this trauma, and you're giving me some string?’

PONE-D-25-43207R1

Dear Dr. Whiffin,

We’re pleased to inform you that your manuscript has been judged scientifically suitable for publication and will be formally accepted for publication once it meets all outstanding technical requirements.

Kind regards,

Kamalakar Surineni, MD, MPH

Guest Editor

PLOS One

Additional Editor Comments (optional):

Reviewers' comments:

Reviewer's Responses to Questions

**Comments to the Author**

1. If the authors have adequately addressed your comments raised in a previous round of review and you feel that this manuscript is now acceptable for publication, you may indicate that here to bypass the “Comments to the Author” section, enter your conflict of interest statement in the “Confidential to Editor” section, and submit your "Accept" recommendation.

Reviewer #1: All comments have been addressed

Reviewer #2: All comments have been addressed

2. Is the manuscript technically sound, and do the data support the conclusions?

Reviewer #1: Yes

Reviewer #2: Yes

3. Has the statistical analysis been performed appropriately and rigorously? 

Reviewer #1: N/A

Reviewer #2: N/A

4. Have the authors made all data underlying the findings in their manuscript fully available?

Reviewer #1: No

Reviewer #2: No

5. Is the manuscript presented in an intelligible fashion and written in standard English?

Reviewer #1: Yes

Reviewer #2: Yes

6. Review Comments to the Author

Reviewer #1: The authors have provided a thorough and thoughtful revision of the manuscript, and I appreciate the detailed point-by-point responses to the previous review. The revisions have significantly strengthened the manuscript across multiple domains.

In particular, the manuscript now demonstrates:

Improved conceptual clarity, with clearer articulation of the theoretical underpinnings of the Life Threads Approach, including links to narrative identity reconstruction, meaning-making, and post-traumatic growth.

Enhanced methodological transparency, especially regarding recruitment, data collection, and the analytic process consistent with reflexive thematic analysis.

Stronger analytic rigor, with clearer description of theme development and inclusion of representative participant quotations across themes.

A more comprehensive discussion, incorporating relevant literature (e.g., ambiguous loss, family systems perspectives) and acknowledging limitations appropriately.

Clarification of ethical considerations, reflexivity, and implementation boundaries, including the need for skilled facilitation and trauma-informed delivery.

The findings are presented in a coherent and compelling manner, and the conclusions are appropriately cautious and aligned with the qualitative nature of the study. The manuscript makes a meaningful contribution to the literature on family experiences following traumatic brain injury and offers valuable insights into narrative-based supportive interventions.

I have no further substantive concerns and believe the manuscript is suitable for publication in its current form.

Reviewer #2: - In response to question 4: The authors have provided a fully compliant Data Availability Statement that appropriately handles ethical restrictions for sensitive qualitative data.

- The shift from "data saturation" to "data sufficiency" effectively justifies the sample size , and detailing the phishing vetting process strengthens study integrity .

- The manuscript is clear, professional, and well-written.

7. PLOS authors have the option to publish the peer review history of their article (what does this mean?). If published, this will include your full peer review and any attached files.

Reviewer #1: **Yes:** VENKATA VIJAYA K DALAI

Reviewer #2: **Yes:** Ivanshu Jain

---

## [Editor Report · Acceptance letter]

PONE-D-25-43207R1

PLOS One

Dear Dr. Whiffin,

I'm pleased to inform you that your manuscript has been deemed suitable for publication in PLOS One. Congratulations! Your manuscript is now being handed over to our production team.

Kind regards,

on behalf of

Dr. Kamalakar Surineni

Guest Editor

PLOS One